# Video Perception Models for 3D Scene Synthesis

**Rui Huang**[1*] **Guangyao Zhai**[2,4*] **Zuria Bauer**[3] **Marc Pollefeys**[3,5]
**Federico Tombari**[2] **Leonidas Guibas**[6] **Gao Huang**[1†] **Francis Engelmann**[6]

[1]Tsinghua University  [2]Technical University of Munich  [3]ETH Zurich
[4]Munich Center for Machine Learning  [5]Microsoft  [6]Stanford University

https://vipscene.github.io

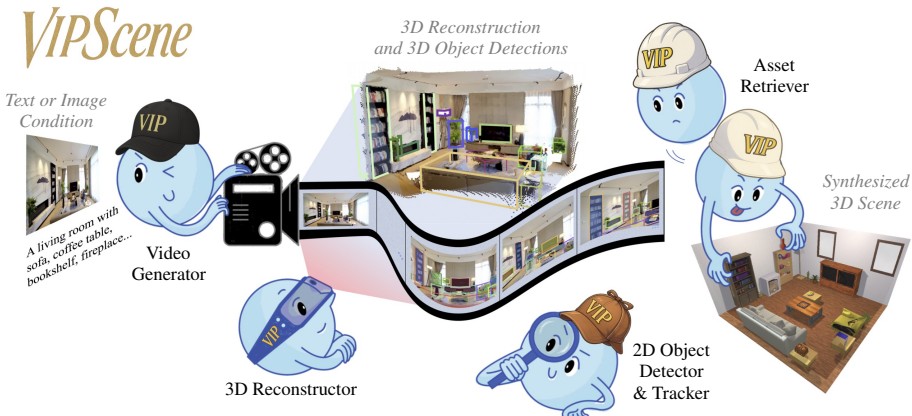

Figure 1: **Schematic of VIPSCENE**. Conditioned on text or image prompts, VIPSCENE generates scenes by leveraging the commonsense priors of video generation models for scene layout and object placements. From the generated video, we reconstruct the 3D scene and extract individual objects. The final scene is synthesized by replacing detected objects with high-quality 3D assets from an object database.

## Abstract

Automating the expert-dependent and labor-intensive task of 3D scene synthesis would significantly benefit fields such as architectural design, robotics simulation, and virtual reality. Recent approaches to 3D scene synthesis often rely on the commonsense reasoning of large language models (LLMs) or strong visual priors from image generation models. However, current LLMs exhibit limited 3D spatial reasoning, undermining the realism and global coherence of synthesized scenes, while image-generation-based methods often constrain viewpoint control and introduce multi-view inconsistencies. In this work, we present **Vi**deo **P**erception models for 3D **Scene** synthesis (VIPSCENE), a novel framework that exploits the encoded commonsense knowledge of the 3D physical world in video generation models to ensure coherent scene layouts and consistent object placements across views. VIPSCENE accepts both text and image prompts and seamlessly integrates video generation, feedforward 3D reconstruction, and open-vocabulary perception models to semantically and geometrically analyze each object in a scene. This enables flexible scene synthesis with high realism and structural consistency. For a more sufficient evaluation on coherence and plausibility, we further introduce **F**irst-**P**erson **V**iew **Score** (FPVSCORE), utilizing a continuous first-person perspective to capitalize on the reasoning ability of multimodal large language models. Extensive experiments show that VIPSCENE significantly outperforms existing methods and generalizes well across diverse scenarios.

---

*Equal contribution.    †Corresponding author.

39th Conference on Neural Information Processing Systems (NeurIPS 2025).

# 1   Introduction

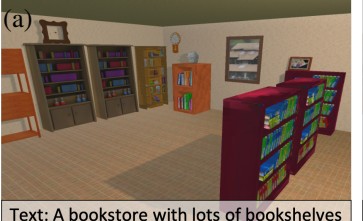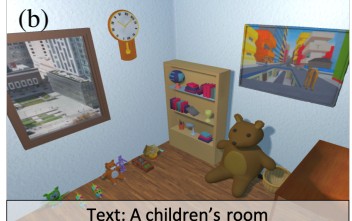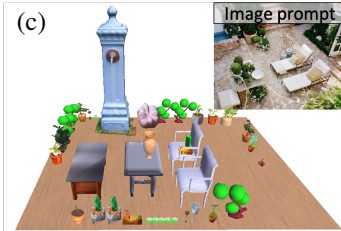

Figure 2: **Diverse Synthesized Environments.** From text prompts, VIPSCENE generates (a) a bookstore and (b) a children's room. From an image prompt (c), VIPSCENE synthesizes a corresponding outdoor scene. Across settings, objects are semantically aligned with the prompts and arranged in physically plausible layouts.

Recent advancements in 3D scene synthesis have sparked significant interest across multiple domains, including gaming [22], augmented reality [1, 50], and robotics [6, 69]. Deep learning-based methods have enabled the automatic generation of 3D scenes; however, they are limited by the insufficient diversity of available datasets. Recent progress in language and image generative models [2, 18, 53–55, 11, 37] has further expanded the possibilities for synthesizing more diverse and plausible 3D scenes. A crucial step in this progress is the generation of spatially coherent layouts, which serve as the foundational structure for building lifelike 3D environments.

Despite impressive progress, existing methods still face notable limitations in generating realistic layouts. One promising direction involves leveraging multimodal large language models (MLLMs) [69, 48, 9, 27, 8, 4, 3]. Among them, pure large language models (LLMs), which typically translate linguistic priors into layout constraints, often produce incomplete spatial specifications and treat scene synthesis as an optimization problem, which can result in the loss of spatial commonsense [69]. Vision-language models (VLMs)[18, 30] attempt to address this issue by incorporating visual context to enhance spatial reasoning. However, their reliance on fixed image viewpoints limits their ability to generalize to 3D layout understanding [48, 34]. Alternatively, image-based approaches [60, 15] adopt specific viewpoints and apply recurrent inpainting strategies to directly exploit visual priors for commonsense layout generation. These methods, however, often suffer from hallucinations and spatial inconsistencies due to viewpoint dependency and the iterative nature.

Moreover, existing automated evaluation protocols often overlook spatial inconsistencies and unrealistic layout distributions. Metrics such as CLIPScore [13] and VQAScore [28], based on VLMs, typically rely on a single top-down view [69, 60]. This perspective can obscure important object details and impede accurate semantic interpretation (see Fig. 7), thus hindering effective assessment of layout coherence. Additionally, top-down views are also likely underrepresented in VLM training data, further reducing their ability to interpret such perspectives. Consequently, these metrics alone can not reliably reflect scene generation quality (see Tab. 2).

In this work, we approach the problem from two complementary perspectives: 3D scene synthesis and evaluation, with a focus on commonsense reasoning and spatial coherence. For scene synthesis, we introduce *Video Perception Models for 3D Scene Synthesis* (VIPSCENE), which leverage rich visual priors from video generation models to capture coherent scene layouts and object placements across views. As shown in Fig. 1, conditioned on either a text or image prompt or both, VIPSCENE first generates an egocentric video of a scene, based on which the feed-forward 3D reconstruction recovers global geometry and open-vocabulary perception segments object instances. The instances enable a large-scale asset retriever to substitute editable 3D assets, followed by a global optimization step to refine object poses, ensuring physical plausibility and avoiding collisions. VIPSCENE enables the generation of realistic, semantically faithful, and spatially coherent 3D scenes (see Fig. 2). Complementing this, we also propose a novel evaluation protocol, the *First-Person View Score* (FPVSCORE), which leverages advanced MLLMs like GPT-4o [18] and Gemini [53, 54] to assess generated scenes. A virtual camera captures 360° first-person views, which are concatenated into visual summaries and analyzed by MLLMs through structured prompts. This approach offers a more interpretable and human-aligned evaluation of layout realism, spatial consistency, and semantic fidelity. Experiments show that VIPSCENE surpasses state-of-the-art baselines across standard metrics, while FPVSCORE outperforms existing metrics and aligns closely with human judgments.

In summary, our contributions are threefold: **(i)** We present VIPSCENE, a novel approach for realistic 3D scene synthesis that leverages video-based commonsense layout understanding, enriched with semantic and geometric cues obtained through consistent 3D reconstruction and perception. **(ii)** We introduce FPVSCORE, a first-person view-based evaluation protocol that leverages MLLMs for more comprehensive, interpretable, and human-aligned assessment of spatial coherence and semantic fidelity. **(iii)** We show that VIPSCENE outperforms state-of-the-art baselines across all metrics, demonstrating the effectiveness of video-grounded priors and a modular decomposition pipeline for generating physically plausible 3D scenes.

## 2 Related Work

**Indoor 3D Scene Synthesis.** Indoor 3D scene synthesis has gained attention for applications in robotics [72, 32] and augmented reality [1, 50]. Existing methods generate scenes from language [51, 69, 48, 39, 10], graph-based instructions [71, 73, 26], or images [17, 60], producing either object-level layouts [71, 70] or complete mesh scenes [15, 47]. However, models trained from scratch often suffer from dataset biases [71, 73, 67, 51]. Recent work thus leverages foundation models for broader generalization. While large language models (LLMs) can capture inter-object commonsense from text, they often yield incomplete or ambiguous spatial layouts due to the lack of visual grounding [9, 69, 4, 3, 68]. In contrast, vision-language models (VLMs) incorporate visual cues to enhance spatial reasoning, but their reliance on fixed viewpoints limits their ability to infer holistic 3D scene layouts [48, 8]. Image-based approaches [60, 15, 29] attempt to exploit commonsense visual priors for scene layout generation, but those based on single or multi-view images often struggle with limited viewpoint coverage and inconsistent spatial alignment across views. In this work, we leverage large-scale pre-trained video diffusion models [33] to generate long-horizon clips from textual or image conditions, ensuring consistent viewpoints while enriching visual details.

**Video Models.** Building on the success of image synthesis [38], recent research has shifted toward video generation, which introduces challenges like temporal consistency and dynamic content modeling. Early efforts mainly relied on GAN-based frameworks [56, 43] that, while producing plausible results in controlled settings, often suffered from mode collapse and temporal incoherence. Recent approaches have extended diffusion models to the video domain [14, 65], leveraging their robustness to generate temporally coherent sequences. With the rise of Diffusion Transformers (DiT)[35], methods now produce highly consistent photorealistic videos[21, 25, 31, 42, 33, 19], revolutionizing applications in film, robotics, and other downstream tasks. In this work, we employ advanced video models to synthesize scenes with consistent views and broad viewpoint coverage, providing a strong foundation for generating 3D scenes with coherent layouts.

**3D Geometric Reconstruction.** Early approaches primarily focused on Structure from Motion (SfM) techniques, where keypoint detection and matching formed the basis for estimating camera poses and sparse point clouds [44, 45]. More recently, learning-based approaches [57, 49, 52, 58, 64], including DUSt3R [59] and MASt3R [23], aim to eliminate iterative optimization and complex post-processing by directly performing multiple 3D tasks through a feedforward network. These methods typically integrate data-driven priors to mitigate ambiguities that classical techniques encounter, demonstrating strong generalization capabilities. In this work, we rely on Fast3R[64] for 3D reconstruction, and MASt3R[23] for tracking of objects from the generated video.

## 3 Method

The goal of VIPSCENE is to generate a realistic and physically plausible 3D scene from a user-specified prompt. Formally, given an image- or text-prompt, VIPSCENE generates a 3D scene $S = \{o_1, \ldots, o_N\}$, where each object $o_i = (c_i, s_i, l_i, \theta_i)$ is represented by its category $c_i \in \mathcal{C}$, size $s_i \in \mathbb{R}^3$, position $l_i \in \mathbb{R}^3$, and orientation $o_i \in \mathbb{R}$ around the gravity axis. Conditioned on the prompt, VIPSCENE first generates a first-person view video of a 3D indoor scene, leveraging commonsense knowledge on scene layout as well as object placements embedded in the video generation model. From this video, we employ recent 3D reconstruction and visual perception models to recover the full 3D scene geometry and decompose it into its individual objects (Sec. 3.1). The scene is then recomposed by retrieving the most similar 3D assets from an object database. To ensure physical plausibility and resolve collisions, an additional optimization step refines object placements (Sec. 3.2). The overall framework is illustrated in Fig. 1, and a detailed algorithm is included in Algorithm 1. In summary, VIPSCENE effectively integrates commonsense knowledge into generated 3D scenes, capturing both the input prompt semantics and the physical plausibility of spatial layouts.

## 3.1 Scene Understanding

**Scene Reconstruction.** Given an input prompt, a conditional video generator (Cosmos [33]) produces a high-fidelity video. For 3D reconstruction, we first sample frames at 2 fps, yielding $\{I_1, \ldots, I_T\}$, where each frame $I_j \in \mathbb{R}^{3 \times H \times W}$ for $j = 1, \ldots, T$, representing diverse views of the scene. Trained on web-scale videos, the generator captures commonsense layout priors and spatial relationships, naturally extending the perceptual field beyond image-based methods like Architect [60]. Next, we process all unposed frames in parallel using a multi-view 3D reconstruction method to produce the 3D scene reconstruction $R$, implemented with Fast3R [64]. For metric 3D reconstruction, we estimate metric depth for each frame using the monocular predictor UniDepth [36], and rescale the reconstructed scene accordingly.

**Object Detection.** Next, we aim to detect all 3D objects in the reconstructed scene $R$. While off-the-shelf 3D object detectors are a natural choice, we found them to perform poorly on the noisy reconstructed point cloud $R$, leading to inaccurate object categorization and size estimation, as shown in Fig. 8. Instead, we adopt an image-based approach. Specifically, we apply Grounded-SAM [41] to detect and segment objects of interest independently in each frame, and then use MASt3R [23] to track and associate 2D detections across frames using its strong multi-view pixel-correspondence estimation capabilities. In this way, we can assign a unique identifier $i$ to each object in the 3D scene. Specifically, for an object $i$ in frame $t$, we store its binary 2D object mask $M_t^i$. We then use the masks across all views, $M_1^i, \ldots, M_T^i$, to extract the corresponding points from the reconstructed point cloud $R$. Given the high degree of noise in $R$, we propose an adaptive erosion scheme that filters out artifacts while preserving the object geometry. Specifically, we apply morphological erosion to each binary object mask $M$ to suppress edge noise, with the erosion strength scaled by object size: larger objects undergo more aggressive denoising, while smaller objects are subject to gentler erosion. This allows us to obtain a clean and accurate point cloud $P_i$ for each object $i$.

## 3.2 Scene Assembly

**3D Asset Retrieval.** In this stage, the goal is to replace the object point clouds $P_i$ with actual 3D assets. Towards that end, we pick the most similar asset from a large-scale object database based on estimated object properties. Specifically, we extend beyond prior approaches (*e.g.*, Holodeck [69]) that rely primarily on visual similarity, textual relevance, and size similarity. Instead, we additionally adopt a point cloud registration-based retrieval strategy to identify the most suitable asset candidates. Given an object point cloud $P_i$, we first estimate its orientation $\theta_i$. We apply Principal Component Analysis (PCA) to compute the principal axes and use the direction of greatest variance to approximate the orientation. A tight bounding box is then aligned with $\theta_i$, from which position $l_i$ and size $s_i$ are derived. As PCA cannot distinguish between 0 and $\pi$, each object yields two symmetric poses. For each candidate asset, we compute a rigid transformation $\mathbf{T} = [\mathbf{R}|\mathbf{t}]$, with rotation $\mathbf{R} \in \mathbb{R}^{3 \times 3}$ and translation $\mathbf{t} \in \mathbb{R}^3$, by minimizing:

$$[\mathbf{R}^*, \mathbf{t}^*] = \underset{\mathbf{R}, \mathbf{t}}{\operatorname{argmin}} \sum_{q \in Q_j} \left( \min_{p \in P_i} \|\mathbf{R}q + \mathbf{t} - p\|^2 \right) + I_{\text{SO}(3)}(\mathbf{R}), \tag{1}$$

where $p$ is a point in the object's point cloud $P_i$ and $q$ is the closest point in the candidate asset $Q_j$. The term $I_{\text{SO}(3)}(\mathbf{R})$ ensures that the rotation matrix $\mathbf{R}$ remains within the special orthogonal group SO(3) as defined in [74]. To solve this optimization problem, we use the Iterative Closest Point (ICP), initialized with the estimated poses. To identify the most suitable asset for each object, we select the candidate with the lowest root mean square error (RMSE), resulting in a scene composed of geometrically well-aligned assets. This step ensures accurate geometric alignment of the assets.

**Object Pose Refinement.** To address potential collisions caused by size mismatches between retrieved assets and objects, we introduce an optimization step to refine object placements in a physically plausible manner. This ensures that objects avoid overlap, stay within room boundaries (if specified), and remain close to their initial position. We define the total loss as a weighted sum of three losses, the position loss $\mathcal{L}_p$, the overlap loss $\mathcal{L}_o$, and the optional boundary loss $\mathcal{L}_b$:

$$\mathcal{L}_{\text{total}} = \mathcal{L}_p + \lambda_o \mathcal{L}_o + \lambda_b \mathcal{L}_b, \tag{2}$$

where $\lambda_o$ and $\lambda_b$ are weighting parameters, and the individual loss terms are defined as:

$$\mathcal{L}_p = \sum_{i=1}^{n} |l_i - l_i^{\text{orig}}|_2^2, \quad \mathcal{L}_o = \sum_{i \neq j} \text{Area}\left(\text{BBox}_i \cap \text{BBox}_j\right), \quad \mathcal{L}_b = \sum_{i=1}^{N} \text{Area}\left(\text{BBox}_i \setminus \text{Room}\right).$$

The position loss $\mathcal{L}_p$ encourages minimal deviation from the objects' original locations $l$. The overlap loss $\mathcal{L}_o$ penalizes intersecting object pairs based on the area of their bounding boxes. The boundary loss $\mathcal{L}_b$ penalizes any part of an object that lies outside the room bounds, if provided. During optimization, object positions are iteratively updated along the gradient of $\mathcal{L}_{\text{total}}$ until convergence. The process terminates once overlap and boundary violations are eliminated or when no significant improvements are observed. This results in a collision-free, spatially coherent scene layout.

## 4   First-Person View Score – FPVSCORE

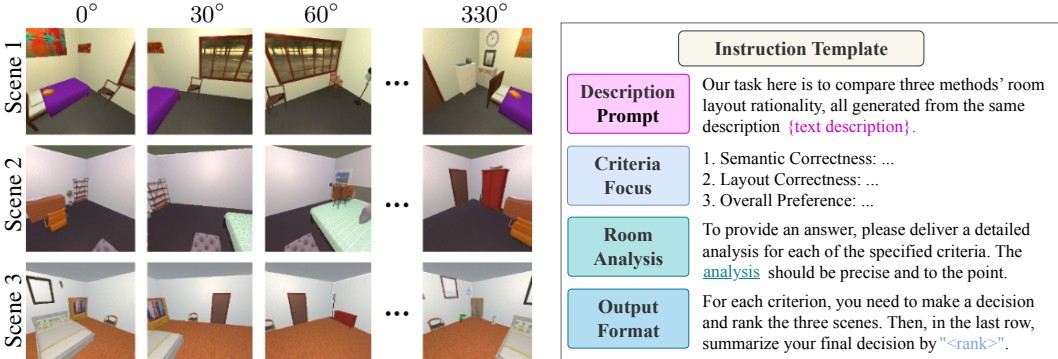

Figure 3: **Illustration of First-Person View Score.** Rather than relying on a *single top-down view*, our metric uses a *sequence of first-person view* images for each generated scene *(left)*. A multimodal language model (MLLM) then analyzes and ranks the sequences based on multiple evaluation criteria *(right)*.

User studies remain the gold standard for evaluating the quality of generated 3D scenes, as they accurately reflect human judgment. However, they are expensive, time-consuming, and difficult to scale. Recent works [69, 60] have explored *automated* evaluation using vision-language models (VLMs) like CLIP [40] and BLIP [24], which assess how well a *top-down* rendering aligns with a given text prompt. For example, CLIPScore[13] computes the cosine similarity between the text encoding of the prompt and the image encoding of the rendered image.

Yet, top-down views may be underrepresented in the training distributions of VLMs, potentially limiting their ability to interpret such inputs accurately. Capturing the full scene layout, including fine-grained geometry and object semantics, in a single image embedding is inherently difficult. These views may also obscure key visual details, further reducing alignment fidelity. As a result, the scores produced by these metrics are often unreliable and hard to interpret meaningfully.

Instead, we introduce an alternative evaluation protocol, called FPVSCORE, that uses multiple *first-person* views, which better reflect the training distributions of foundation models and offer improved scene coverage. Crucially, rather than using a single similarity metric, we exploit the perceptual and reasoning capabilities of multimodal models such as GPT-4o [18] and Gemini [53, 54] to rank the outputs of 3D scene generation models and to explain the rationale behind each ranking.

**Metric Details and Prompt Design.**   Fig. 3 illustrates our first-person view metric. For each scene, a virtual camera is placed at the center and simulates a 360-degree rotation, rendering frames at 30-degree intervals to ensure comprehensive coverage and sufficient overlap. These frames are horizontally concatenated to form a compact visual summary. To enable consistent comparison across methods, we stack the summaries for all scenes and input them into the multimodal large language models (MLLMs) simultaneously, avoiding inconsistencies that could arise when evaluating them in isolation. Building on prior work in multimodal model-based 3D object assessment [61], we design a structured prompt to enable comparative evaluation of scene generations. As illustrated in Fig. 3, the prompt comprises three key elements: (1) task-specific instructions defining the multi-scene comparison goal, (2) a clear list of evaluation criteria, and (3) formatting guidelines to ensure consistent output. The prompt guides the model to assess scenes along dimensions such as semantic correctness, layout accuracy, and overall coherence. We further instruct the model to justify its ratings with brief explanations, enabling verification of its reasoning and increasing trust in the evaluation. This design facilitates a more holistic, interpretable, and scalable evaluation protocol for 3D scene generation, addressing the limitations of traditional top-down metrics by aligning more closely with human-like reasoning patterns (see Tab. 2).

# 5 Experiments

**Experimental Details.** We utilize Cosmos [33] for video generation and adopt Fast3R [64] for 3D reconstruction. For open-vocabulary segmentation, we employ Grounded-SAM [41], and UniDepth [36] is applied for monocular depth estimation. For comparison with baselines, we follow prior work [69] and evaluate on four types of scenes *living room*, *bedroom*, *bathroom*, and *kitchen*. We ask GPT-4o [18] to produce 25 text prompts for each room type. Each prompt consists of a description of a room type and the desired items. Based on these prompts, we generate 100 rooms using each method under evaluation. We set $\lambda_o = \lambda_b = 10$. Consistent with prior work, Holodeck [69], we retrieve 3D models from a high-quality subset of Objaverse [7] to ensure realistic and diverse object representations in the scene. Please refer to the appendix for more details.

**Baselines.** We compare our method with the most recent state-of-the-art approaches for 3D scene synthesis: Holodeck [69] is a comprehensive system that integrates LLM-based scene generation with optimization steps to jointly produce room layouts and object placements. Architect [60] is a generative framework that creates interactive 3D scenes through diffusion-based 2D inpainting, relying on visual priors extracted from single images.

**Metrics.** To evaluate the quality of generated scenes, we report scores using the proposed first-person view metric FPVSCORE (Sec.4) and other automatic metrics used in prior work [60]: (1) CLIPScore [13], which measures image-text similarity via CLIP embeddings; (2) BLIPScore, which evaluates image-caption alignment using the matching head of BLIPv2 [24]; (3) VQAScore [28], which uses a visual question answering model to score how likely an image depicts the given caption; and (4) GPT-4o Ranking [18], which prompts GPT-4o to rank top-down rendered views.

**User Study.** We conduct a user study to compare scenes generated by our method against baseline approaches. Participants are shown a 360-degree video captured from the center of each scene, along with a top-down rendered image, allowing them to assess both global structure and fine details. Thirty graduate students rated the scenes on a 3-point scale (1 = lowest, 3 = highest) across three criteria: Prompt Adherence (PA), *"To what extent does the generated scene align with the input prompt?"*, Layout Correctness (LC), *"Are the object placements physically plausible and functionally sensible?"*, and Overall Preference (OP).

## 5.1 Quantitative Results

Tab.1 presents quantitative results comparing our VIPSCENE to the Holodeck[69] and Architect [60] baselines. We report 2D image-based metrics using both first-person and top-down views, along with user study outcomes. VIPSCENE outperforms both baselines across all metrics. The user study, our most reliable evaluation, indicates that VIPSCENE better captures prompt semantics, produces more realistic scene layouts, and is the overall preferred method. This trend is also reflected in our proposed first-person view metric across three different VLMs (Gemini 2.0, GPT-4o, GPT 4.1). Top-down view scores offer a much less clear picture. According to these metrics, performance across methods is nearly indistinguishable, a finding not supported by the more reliable user study. This supports our intuition that top-down view metrics are poorly suited for evaluating 3D scene generation; we examine this further in Sec. 5.3. Another notable finding from our study is that Architect generally underperforms compared to Holodeck on the 100 generated scenes. This contrasts with the results reported in [60], but is partially supported by the qualitative examples in Fig. 4, which illustrate that Architect often produces unusual and sometimes impractical object placements.

| Method | First-Person View Scores | | | User Study | | | Top-Down View Scores | | | |
|---|---|---|---|---|---|---|---|---|---|---|
| | Gemini 2.0↑ | GPT-4o↑ | GPT 4.1↑ | PA↑ | LC↑ | OP↑ | CLIP↑ | BLIP↑ | VQAScore↑ | GPT-4o↑ |
| Holodeck [69] | 1.92 | 2.02 | 1.94 | 2.31 | 2.06 | 2.05 | 29.17 | 51.27 | 81.43 | 1.98 |
| Architect [60] | 1.77 | 1.62 | 1.76 | 2.05 | 1.94 | 1.98 | 29.95 | 49.72 | 78.34 | 1.90 |
| VIPSCENE (Ours) | **2.32** | **2.45** | **2.43** | **2.52** | **2.51** | **2.39** | **29.98** | **54.36** | **82.13** | **2.12** |

Table 1: **First-Person View, User Study, and Top-Down View Scores.** We report scores across different evaluation metrics. For First-Person View Scores and GPT-4o-based metrics, we report average ranking where 3 is the best and 1 is the worst. Prompt Adherence (PA), Layout Correctness (LC), Overall Preference (OP).

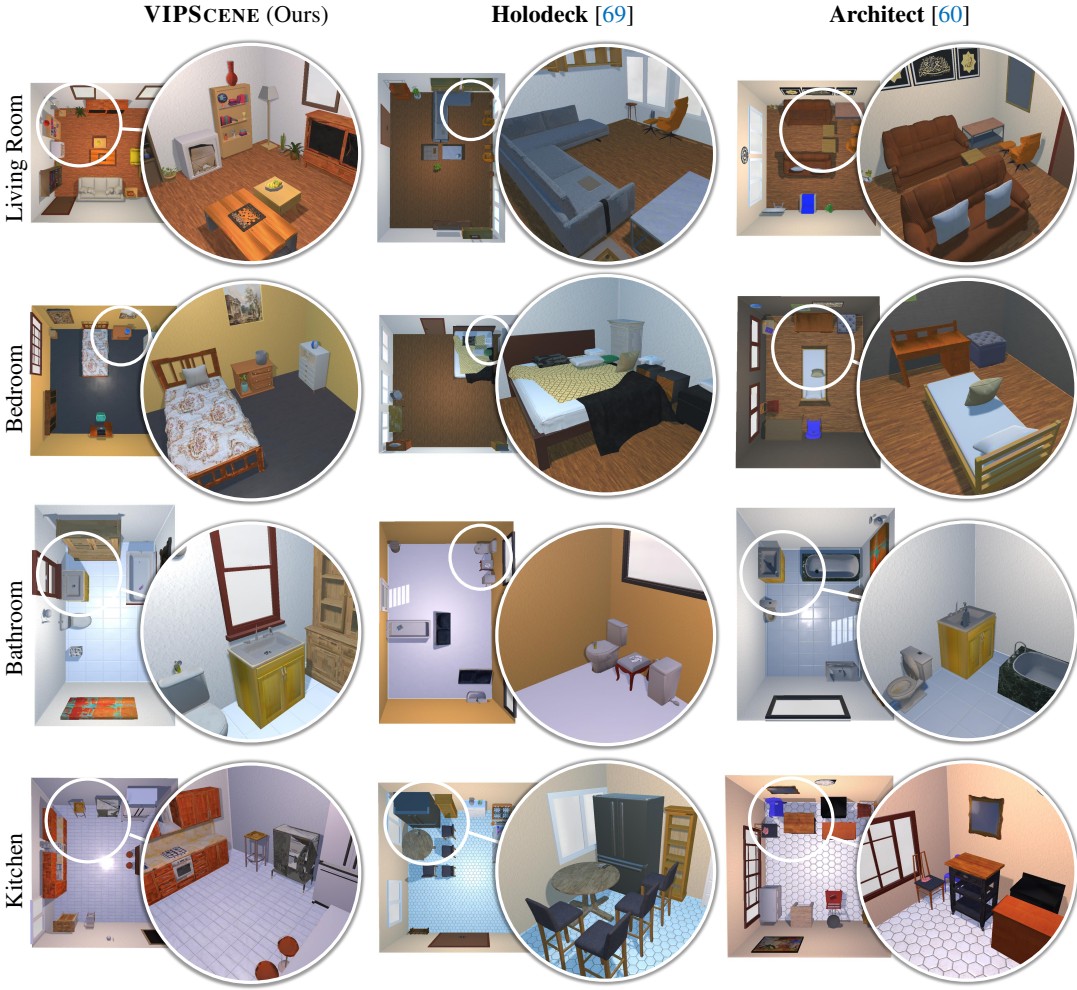

Figure 4: **Qualitative Results.** We present top-down and close-up views of our VIPSCENE, comparing it against Holodeck [69] and Architect [60] *(columns)* across four room types *(rows)*. In the figures, Holodeck clearly leaves large areas unused while over-cluttering others, whereas Architect produces implausible arrangements that are impractical and rarely seen in real environments. VIPSCENE generates room layouts that are overall more realistic and natural.

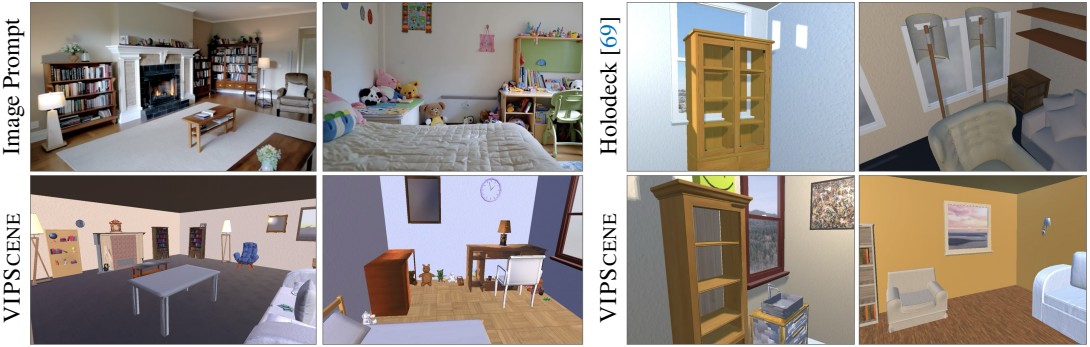

Figure 5: **Image Prompting and Scene Realism.** *Left:* Examples of image-based prompting: given an input image, VIPSCENE generates a video and reconstructs a full 3D scene. Note that based on the information from the first frame, the generated video can plausibly hallucinate objects beyond the original field of view, such as the white sofa near the observer in the living room or the side window in the bedroom. *Right:* Additional detailed comparison of scene layouts. Unlike Holodeck [69] *(top)*, which handles object placement and window layout separately, our method *(bottom)* jointly reasons about their spatial relationships, avoiding window occlusions and yielding more coherent, realistic arrangements.

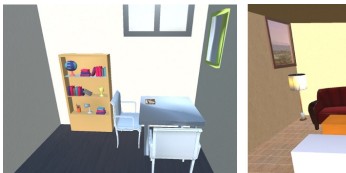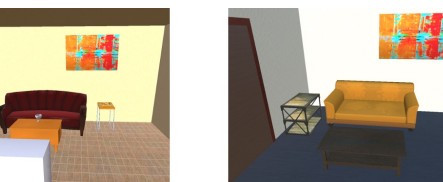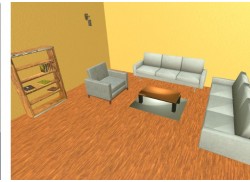

Figure 6: **Results from Complex Inputs.** *Left:* Our model can effectively generate coherent videos and corresponding 3D scenes from detailed spatial prompts, such as "a desk placed under a window with a bookshelf to its left" and "a modern living room with a red sofa facing a fireplace, a coffee table in between, and a floor lamp placed diagonally behind the sofa." *Right:* To further enhance realism, we utilize CLIP features from cropped object regions during retrieval. This improves the alignment of textures and materials with input prompts like "a yellow sofa in the center of the room, with a dark rectangular coffee table in front and a metal shelf next to the sofa" and "the room has two light-colored sofas, an upholstered armchair nearby, a modern brown coffee table in the center, and a wooden bookshelf against the wall."

## 5.2 Qualitative Results

We present qualitative results in Fig. 4 to 6. For additional results, including those from multimodal inputs, please refer to the appendix. Fig. 4 specifically compares our VIPSCENE with Holodeck [69] and Architect [60], displaying randomly selected outputs from text prompts for living room, bedroom, bathroom, and kitchen scenes. Holodeck, relying solely on LLMs for spatial constraints and object relationships, often produces implausible absolute placements despite semantically correct relative pairings (*e.g.*, chairs around a table). This is because LLMs exhibit limited 3D spatial reasoning, leading to objects being too close for passage while large room areas remain unused. Architect employs multi-view image diffusion and inpainting to hierarchically populate corner views, yielding a more natural mix of object scales. However, its resulting layouts are often incoherent, with issues like multiple sofas facing the same direction and implausible furniture placements. This incoherence likely stems from view inconsistencies and inpainting limitations.

Fig. 5 *(left)* shows scenes generated from image inputs by replacing the text-to-video model with an image-to-video model [21], while keeping the rest of the pipeline unchanged. The generated scenes demonstrate strong realism and coherence, with objects arranged in a manner consistent with the overall video context. It is worth noting that, relying solely on the first frame, the generated video can reasonably infer objects beyond the original field of view, such as the white sofa near the observer in the living room and the side window in the bedroom. Fig. 5 *(right)* showcases how existing methods overlook the joint spatial relationship between objects and windows, whereas our method explicitly models their relative placement. By leveraging spatial patterns in video data, our approach achieves greater logical consistency and enhanced visual plausibility.

We further evaluate our model using more complex textual descriptions, as illustrated in Fig. 6. The video generation model successfully interprets these detailed spatial instructions, producing coherent and realistic layouts that reflect the specified relationships among objects. These generated videos are subsequently used to reconstruct the corresponding 3D scenes, as shown in Fig. 6 *(left)*. Our initial retrieval strategy primarily focused on object category and point cloud geometry to ensure overall scene plausibility. To better align retrieved assets with the textures and materials implied by the input prompt, we conduct additional experiments leveraging CLIP image features extracted from cropped object regions in the video frames. This enhancement enables the system to retrieve assets that more accurately reflect the semantic and visual cues specified in the text, as demonstrated in Fig. 6 *(right)*.

## 5.3 Metrics Evaluation

A key challenge in evaluating generated 3D scenes is the lack of a suitable metric. Prior work [69, 60] typically uses 2D image-based metrics such as CLIPScore [13], BLIP-Score [24], or VQAScore [28]; however, motivated by the surprisingly similar scores across methods observed in Sec. 5.1 (Tab. 1), we investigate how well these metrics actually align with human preferences. Specifically, we compute Kendall's $\tau$ correlation [20] between metric-generated scores and reference scores from hu-

| Metrics | $\tau$ ($\uparrow$) |
|---|---|
| CLIPScore (single top-down) [13] | 0.06 |
| BLIPScore (single top-down) [24] | 0.07 |
| VQAScore (single top-down) [28] | 0.13 |
| GPT-4o (single top-down) | 0.27 |
| GPT-4o (FPVSCORE, first-person views) | **0.39** |

Table 2: **Metrics Evaluation.** How well do automated metrics agree with human ratings? Scores are Kendall's $\tau$ correlations. Perfect agreement is $\tau = 1$, no association is $\tau = 0$, perfect disagreement is $\tau = -1$.

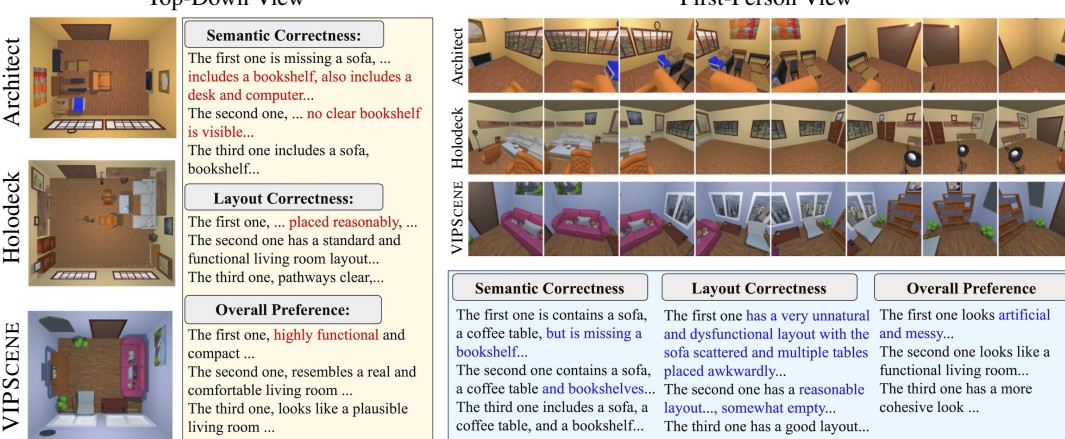

Input Text: "A living room with a coffee table, sofa, and bookshelf."

Figure 7: **GPT-4o Output Analysis.** Exemplary GPT-4o output: top-down view *(left)* and first-person view *(right)*. *Red text* highlights implausible results; *blue text* marks reasonable ones.

man evaluators. Tab. 2 reports the correlations, showing how closely each metric's predictions align with human judgments. Both CLIPScore and BLIPScore exhibit almost no association with human preference, suggesting that these metrics are not suitable for automated evaluation. Although overall correlations are relatively low (reflecting the subjectivity of the task and variability in human judgments), our proposed first-person view metric, FPVSCORE, shows the strongest alignment with human preferences, suggesting that first-person views provide richer semantic cues leading to more reliable evaluations. Fig. 7 illustrates a comparison of the same model's reasoning for top-down and first-person views. Top-down views frequently obscure key object details, hindering semantic understanding and impairing layout evaluation.

| Model | $\tau$ |
|---|---|
| Gemini 2.0 | 0.61 |
| GPT-4o | 0.74 |
| GPT-4.1 | 0.71 |

(a) $\tau$ vs. MLLMs

| Model | $\tau$ |
|---|---|
| Gemini 2.0 & GPT-4o | 0.54 |
| Gemini 2.0 & GPT-4.1 | 0.47 |
| GPT-4.1 & GPT-4o | 0.60 |

(b) Agreement across MLLMs

| Prompt | $\tau$ |
|---|---|
| w/o criteria and analysis | 0.31 |
| Ours | 0.39 |

(c) Prompt design

Table 3: **FPVScore Consistency Evaluation using Kendall's $\tau$ correlation:** Single-model stability, inter-model agreement, and prompt variants.

We further analyze the consistency of FPVSCORE to validate FPVScore's reliability, focusing on three aspects: *(i) Consistency Across Tries.* We repeatedly query each model and compute the average Kendall's $\tau$ correlation. Results (Tab. 3a) show MLLMs yield stable outputs, with GPT-4o demonstrating the highest consistency. *(ii) Consistency Across Models.* We assess agreement among different MLLMs by measuring pairwise Kendall's $\tau$ correlations of their scene rankings. Results (Tab. 3b) indicate models exhibit similar relative judgments. *(iii) Prompt Design.* Removing specific prompt instructions (Tab. 3c) reduces Kendall's $\tau$ correlation, suggesting structured prompts enhance human alignment beyond MLLMs' inherent reasoning.

## 5.4 Ablation Study

In this section, we present ablation studies on a randomly selected subset of scenes to evaluate the contribution of individual model components. For each variant with one component removed, we compare it against the full VIPSCENE model in a user preference test. Fig. 8 reports the percentage of times VIPSCENE was favored over the model variant.

**2D *vs.* 3D Perception Models.** After reconstructing the scene's point cloud, one straightforward approach is to directly apply a 3D perception model for scene decomposition. However, in practice, we observe that using a 3D model, such as Mask3D [46], yields suboptimal results. The model suffers from significant classification errors across various object categories, and its segmentation masks are often inaccurate, leading to poor scene composition. We believe this issue stems from the higher levels of noise present in the point cloud generated by the video generation model, which differs significantly from the distribution of the

| Frames | FPVSCORE |
|---|---|
| 5 | 2.02 |
| 10 | 2.50 |
| 20 | 2.48 |

Table 4: **Ablation on Frame Quantity in Perception.**

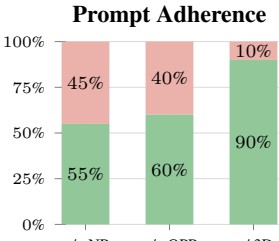
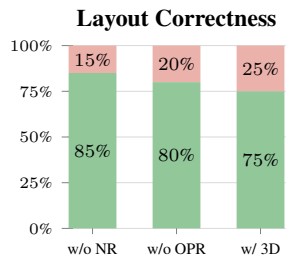
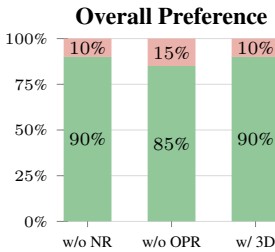

Figure 8: **Ablation Study.** Win ratio of VIPSCENE versus variants measured by prompt adherence (PA), layout correctness (LC), and overall performance (OP). Variants include the original model without noise reduction (NR), without object pose refinement (OPR), and using a 3D instead of a 2D object detector. A 50% win ratio indicates equal performance, while 100% means the full model always outperforms the variant.

training data used for the 3D perception model. In contrast, our approach begins with 2D perception models, which are more robust in accurately identifying object categories in images. This is further enhanced by point cloud denoising techniques, which reduce the noise in the reconstructed point cloud. As a result, our method achieves more reliable and precise scene decompositions, ultimately leading to higher-quality scene synthesis. To assess 2D perception efficiency, we analyze the trade-off between frame count and accuracy. As shown in Tab. 4, enlarging the temporal window improves performance but quickly saturates. In practice, sampling at 2 fps and using ~10 frames strikes a favorable balance between cost and accuracy.

**Noise Reduction (NR).** Due to potential motion blur and other artifacts in the generated video, as well as the 3D reconstruction model itself, the reconstructed point clouds may exhibit significant noise, especially at the border of object masks, i.e., at depth discontinuities. The adaptive erosion method we introduce effectively filters out these artifacts while preserving the integrity of the target objects, facilitating more precise scene decomposition and improving the asset retrieval process. Beyond the full-module ablation, we also compare fixed-kernel variants. The adaptive setting consistently outperforms others (Tab. 5).

| Erosion Method | FPVSCORE |
|---|---|
| Small Kernel | 2.30 |
| Large Kernel | 2.05 |
| Adaptive | 2.67 |

Table 5: **Ablation on Adaptive Erosion.**

**Object Pose Refinement (OPR).** As demonstrated in Fig. 8, this refinement step effectively mitigates object collisions that may arise from size mismatches between the retrieved assets and the target objects. Following this step, users observe enhanced scene coherence, where objects are maintained close to their original placements while avoiding collisions with each other. This results in a more realistic and visually pleasing scene. We further conduct additional ablation experiments to better isolate the impact of each component in the object pose refinement loss. The results are summarized in Tab. 6. It demonstrates that each loss term makes a meaningful contribution to enhancing the physical realism and stability of the final scenes.

| Loss Variant | Win ratio (%) |
|---|---|
| w/o position loss $\mathcal{L}_p$ | 60 |
| w/o overlap loss $\mathcal{L}_o$ | 85 |
| w/o boundary loss $\mathcal{L}_b$ | 75 |

Table 6: **Ablation on Refinement Loss.**

## 6 Conclusion

In this work, we present VIPSCENE, a novel framework that leverages video perception models for 3D scene synthesis. By integrating video generation, 3D reconstruction, open-vocabulary object detection and tracking, as well as 3D asset retrieval, VIPSCENE bridges the gap between multimodal prompts and coherent, editable 3D scenes. Our method addresses existing challenges in spatial reasoning and multi-view consistency that limit current approaches based on language and image generation models. Furthermore, we introduce a new evaluation metric FPVSCORE that aligns better with human judgment of 3D scenes, offering a more reliable measure of semantic and spatial correctness in generated scenes. Through extensive experiments and user studies, VIPSCENE demonstrates superior performance across diverse scene types, both qualitatively and quantitatively. The obtained results highlight the importance of commonsense knowledge from video data and point toward a promising direction for future research in realistic and interpretable 3D scene generation.

**Acknowledgment** This work is supported in part by the National Key R&D Program of China under Grant 2024YFB4708200, the National Natural Science Foundation of China under Grants U24B20173 and 42327901, and the Scientific Research Innovation Capability Support Project for Young Faculty under Grant ZYGXQNJSKYCXNLZCXM-I20.

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

# Appendix

## A  Detailed Algorithm

This section provides a comprehensive breakdown of the proposed algorithm, outlining its modules, their interactions, and the data representations used throughout the process.

---

**Algorithm 1:** VIPSCENE: Prompt-to-Scene Generation

---

**Input:** Prompt $\pi$ (text / image), video generator $\mathcal{G}$, reconstructor $\mathcal{R}$ (Fast3R), monocular depth $\mathcal{D}$ (UniDepth), detector $\mathcal{S}$ (Grounded-SAM), tracker $\mathcal{T}$ (MASt3R), assets $\mathcal{A}$, weights $\lambda_o, \lambda_b$
**Output:** Final metrically scaled collision-free scene $S = \{o_i\}_{i=1}^N$, where each object $o_i = (c_i, s_i, l_i, \theta_i)$ is represented by its category $c_i \in \mathcal{C}$, size $s_i \in \mathbb{R}^3$, position $l_i \in \mathbb{R}^3$, and orientation $\theta_i \in \mathbb{R}$ around the gravity axis.

1 **(A) Prompt → Video frames**
2 $V \leftarrow \mathcal{G}(\pi)$
3 $\{I_t\}_{t=1}^T \leftarrow$ Sample frames from $V$ with fps = 2
4 **(B) Video → Metric 3D reconstruction**
5 $R \leftarrow \mathcal{R}(\{I_t\}_{t=1}^T)$       `// Globally consistent 3D (unposed inputs)`
6 $\{D_t\}_{t=1}^T \leftarrow \mathcal{D}(\{I_t\}_{t=1}^T)$       `// Metric depths`
7 $R \leftarrow$ Rescale $R$ with metric$\{D_t\}_{t=1}^T$       `// Enforce metric scale`
8 **(C) Scene decomposition & object extraction**
9 **for** $t = 1$ **to** $T$ **do**
10     $\{(c, M_t^{(k)})\}_k \leftarrow \mathcal{S}(I_t)$       `// Per-frame 2D instance masks with categories`
11     $\{M_t^{(k)}\}_k \leftarrow$ AdaptiveErode$(\{M_t^{(k)}\}_k)$       `// Size-aware morphological denoising`
12 $\left\{\{M_t^i\}_{t=1}^T\right\}_{i=1}^N \leftarrow \mathcal{T}(\{I_t, \{M_t^{(k)}\}_k\}_{t=1}^T)$       `// Temporal association / IDs`
13 **for** $i = 1$ **to** $N$ **do**
14     $P_i \leftarrow$ Segment points from $R$ by$\{M_t^i\}_{t=1}^T$       `// Per-object point cloud`
15     $c_i \leftarrow$ Majority label$(\{M_t^i\}_{t=1}^T)$       `// Object category from detections`
16 **(D) 3D asset retrieval & alignment**
17 **for** $i = 1$ **to** $N$ **do**
18     $(s_i, l_i^{\text{init}}, \theta_i^{\text{init}}) \leftarrow$ PCAInit$(P_i)$
19     $\mathcal{C}_i \leftarrow$ Retrieve candidates from $\mathcal{A}$ according to $c_i$ best $\leftarrow \varnothing$, rmse$_{\min} \leftarrow +\infty$
20     **foreach** $Q \in \mathcal{C}_i$ **do**
21        **foreach** $\theta \in \{\theta_i^{init}, \theta_i^{init} + \pi\}$ **do**
22           $(\mathbf{R}^*, \mathbf{t}^*)$, rmse $\leftarrow$ ICPAlign$(P_i, Q; l_i^{\text{init}}, \theta)$       `// Eq. (1),` $\mathbf{R} \in$ SO(3)
23           **if** rmse $<$ rmse$_{\min}$ **then**
24              rmse$_{\min} \leftarrow$ rmse,
25              best $\leftarrow (Q, \mathbf{R}^*, \mathbf{t}^*, \theta)$
26     $(Q_i, \mathbf{R}_i, \mathbf{t}_i, \theta_i) \leftarrow$ best, $l_i \leftarrow \mathbf{t}_i$
27 **(E) Final scene refinement**
28 $l_i^{\text{orig}} \leftarrow l_i \quad \forall i$       `//` $l_i$ `denotes position variables`
29 **repeat**       `// Gradient-based optimization`
30     $\mathcal{L}_p = \sum_{i=1}^N \|l_i - l_i^{\text{orig}}\|_2^2$
31     $\mathcal{L}_o = \sum_{i \neq j}$ Area(BBox$_i(l_i, s_i) \cap$ BBox$_j(l_j, s_j))$
32     $\mathcal{L}_b = \sum_{i=1}^N$ Area(BBox$_i(l_i, s_i) \setminus$ Room)
33     $\mathcal{L}_{\text{total}} \leftarrow \mathcal{L}_p + \lambda_o \mathcal{L}_o + \lambda_b \mathcal{L}_b$
34     $\{l_i\} \leftarrow$ Update$(\{l_i\}, -\eta \nabla_{\{l_i\}} \mathcal{L}_{\text{total}})$ **if** Overlap$(\{BBox_i\}) = 0$ **and** $\Delta\mathcal{L}_{total} < \varepsilon$ **then**
35        **break**
36 **until** *converged*
37 **return** $S = \{(c_i, s_i, l_i, \theta_i)\}_{i=1}^N$

---

# B    Computational Complexity

We have profiled both the inference latency and peak GPU memory consumption of each major stage in our pipeline in Tab. 7. Among all stages, video generation is the most computationally expensive, requiring around 380s per video and 74GB GPU memory on an H100. 3D reconstruction, object detection and tracking, as well as asset retrieval, are significantly more efficient, each taking only a few seconds per scene with much lower memory usage. The total latency for generating a complete scene is 400s, compared to 200s in Holodeck. Although our pipeline is more computationally intensive in the generation stage, it yields significantly higher scene quality, which we believe justifies the trade-off.

Table 7: **Runtime and Memory Footprint per Stage.**

| Stage | Time Cost | Memory Usage |
|---|---|---|
| Video Generation | ~380 s | ~74 GB |
| 3D Reconstruction | ~2 s | ~8 GB |
| Object Detection & Tracking | ~15 s | ~8 GB |
| 3D Asset Retrieval | ~5 s | — |
| Pose Refinement | ~1 s | — |

# C    Additional Qualitative Results

We provide additional qualitative results, including experiments on scene synthesis from multimodal and text-only inputs, as shown in Fig. 9 and Fig. 10. Specifically, our video generation model utilizes both image and text inputs as complementary modalities for 3D scene generation. The input image defines the original field of view, while the accompanying text guides the model to plausibly infer and complete scene elements beyond the visible region. This multimodal setting enhances the flexibility and robustness of the generation process, resulting in more accurate and diverse 3D scenes.

# D    Adaptive Erosion Method

We apply an adaptive erosion strategy to remove noise and retain high-confidence points in the reconstructed point clouds. As shown in Fig. 11, this approach effectively reduces artifacts in the initial per-object point clouds, resulting in cleaner and more coherent geometries. These improvements further benefit the subsequent asset retrieval process and enhance the overall scene quality.

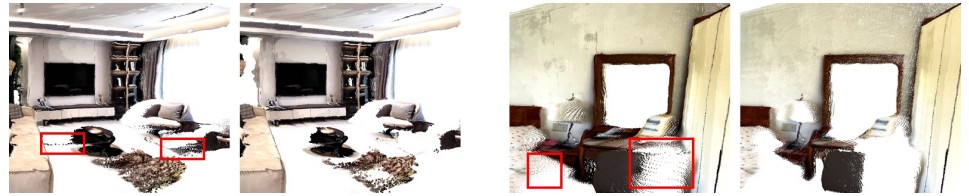

Figure 11: **Before-and-After Visualization of the Adaptive Erosion Strategy.** The method effectively removes noisy points and preserves high-confidence regions, producing cleaner and more consistent geometries.

# E    Failure Cases

We show representative failure cases in Fig. 12. Typical errors include incorrect category recognition by the detector and missing detections caused by occlusion. When objects are dense, the layout optimization is not realistic enough. In the left example, a misclassified fireplace results in duplicated instances, whereas in the right example, the placement of the wooden table is not reasonable enough.

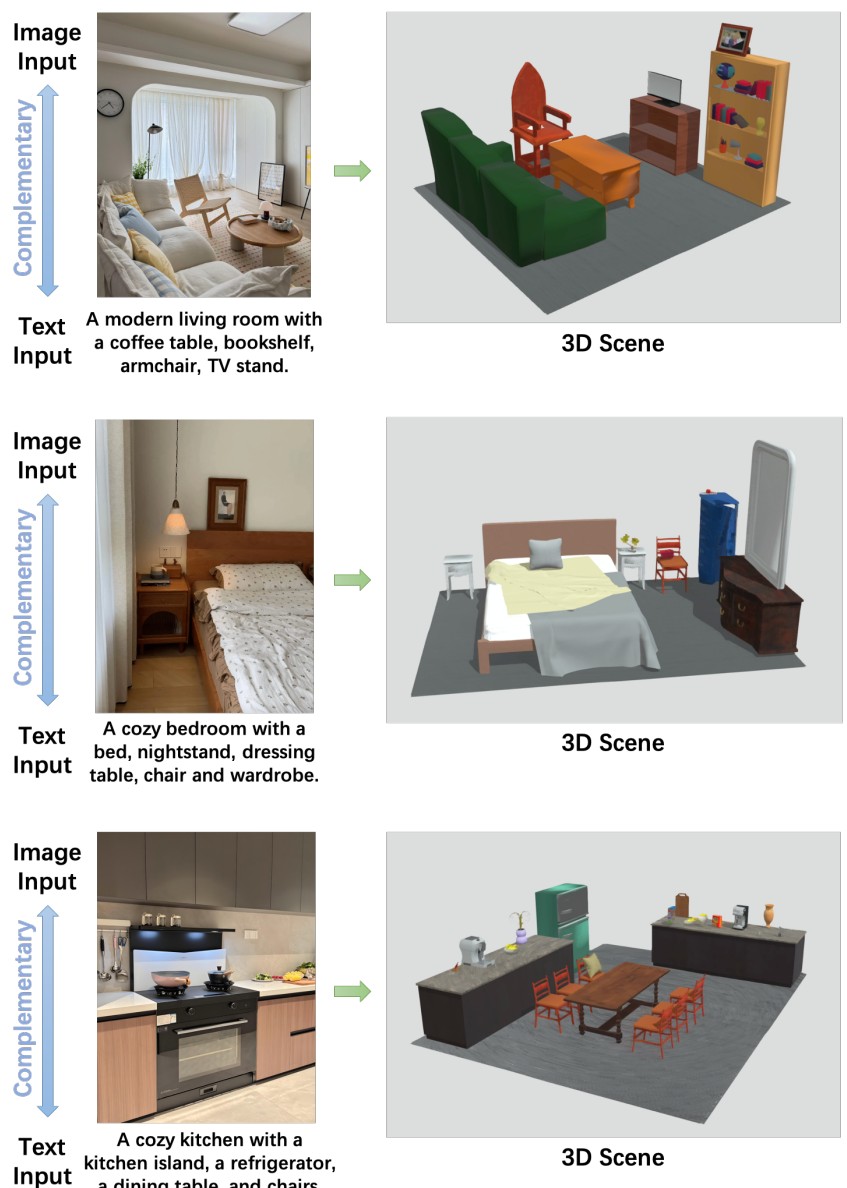

Figure 9: **Qualitative Results from Multimodal Inputs.** The generated video respects the original field of view provided by the input image while leveraging the accompanying text to plausibly infer and complete scene elements beyond the visible area.

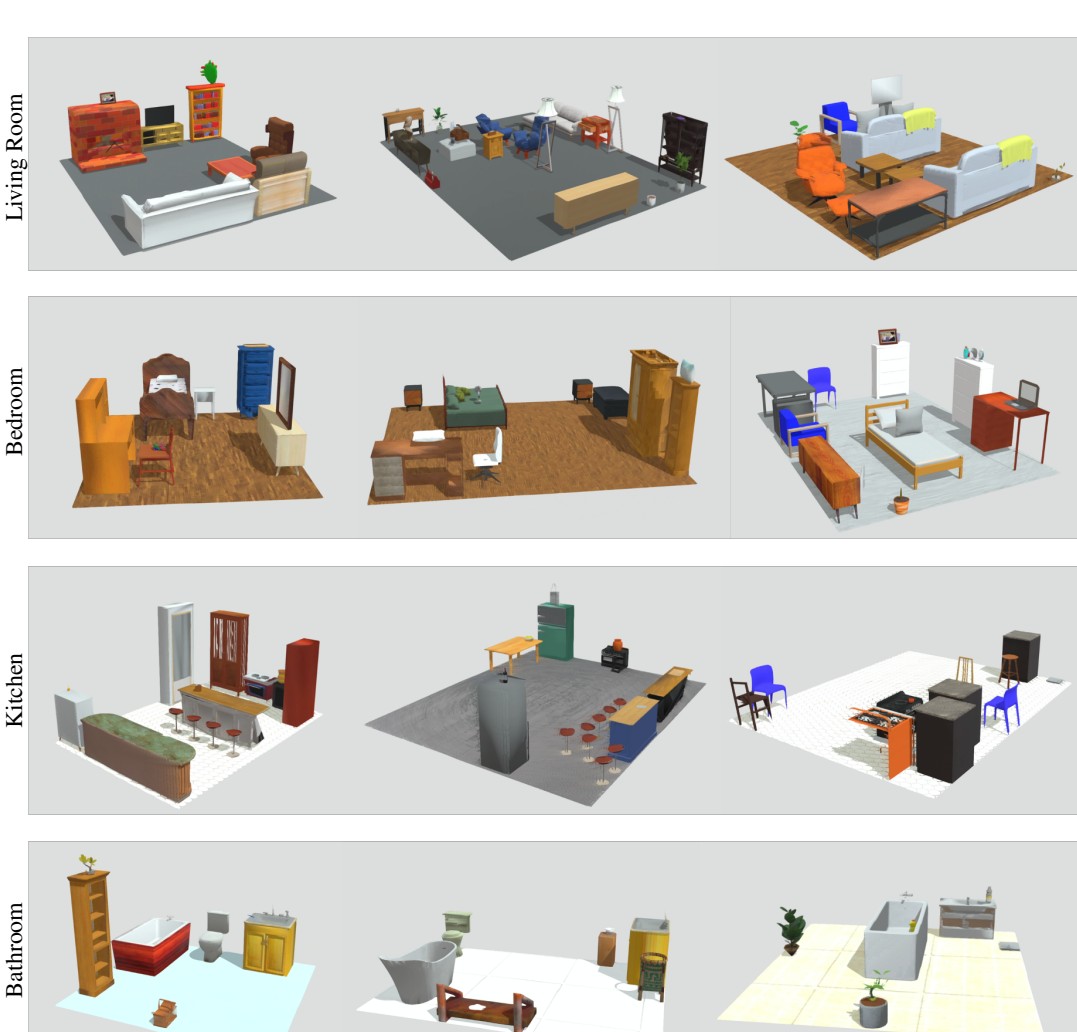

Figure 10: **Additional Qualitative Results.** We present results of VIPSCENE, comparing it against Holodeck [69] and Architect [60] *(columns)* across four room types *(rows)*. For better visibility, ceilings and walls are removed.

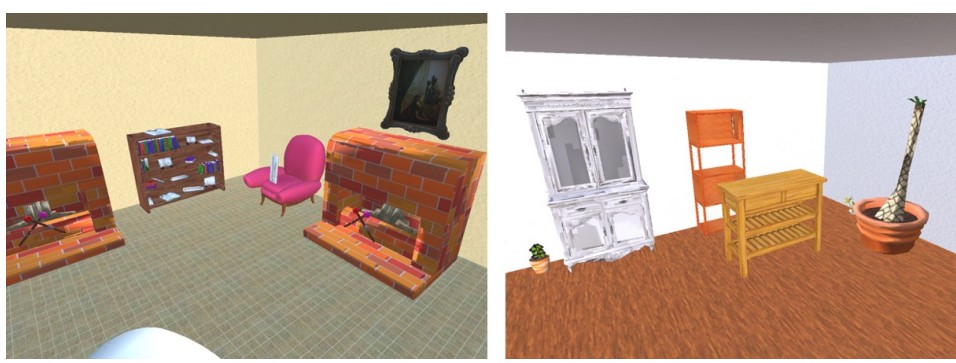

Figure 12: **Failure Cases.** *Left:* Misclassification by the detector results in duplicated fireplaces. *Right:* Inaccurate placement of a wooden table due to layout optimization issues in dense scenes.

# F  Implementation Details

**Rescaling the Scene.**   To refine the overall scale of the reconstructed scene, we estimate depth maps for each view using UniDepth [36], and compare them against the corresponding reconstructed point clouds. For each point, we compute the ratio between the estimated depth and the depth derived from the reconstructed geometry. The global scale factor is then determined as the median of these per-point ratios across all views, providing a robust estimate that mitigates the influence of outliers. This median-based scaling approach ensures consistency across views and improves the alignment of the reconstructed scene with real-world metric dimensions.

**Orientation Estimation of the Object.**   Without loss of generality, we assume that object bounding boxes are aligned with the ground plane. To estimate their orientation, we first determine the ground plane equation. This process begins by extracting the ground point cloud, using a method analogous to object extraction. Specifically, we prompt Grounded-SAM [41] with the label "ground" for outdoor scenes or "floor" for indoor scenes to generate ground masks. These masks are then used to extract the corresponding ground points from the reconstructed scene, and a least-squares fitting is applied to estimate the ground plane. With the ground plane established, each object's point cloud $P_i$ is transformed into a new coordinate system that retains the origin of the original camera coordinate system $C$, but aligns its horizontal plane with the estimated ground plane. The transformed point cloud is then projected onto the ground plane, and Principal Component Analysis (PCA) is applied to identify the principal axes of the point distribution. The direction of greatest variance is taken as an approximation of the object's orientation $\theta_i$. A tight bounding box is subsequently aligned with this estimated direction.

# G  Prompts Details

**Prompts for Scene Synthesis.**   We utilize GPT-4o [18] to generate text prompts for four types of indoor scenes: living room, bedroom, kitchen, and bathroom. Each prompt specifies the room type along with the objects intended to furnish the space. For example, "A bedroom with a large bed, two nightstands, a floor lamp, a wardrobe, and a big window."

> I'm working on an interior design project and would like to generate video scenes of a {room type} using a text-to-video model. Please help me create detailed prompts to feed into the model.
> Guidelines:
> 1. Based on the typical function and layout of a {room type}, list the furniture, appliances, decorations, and other items commonly found in the space. 2. Prompts should describe the room's contents clearly and in detail.
> Example: "A bedroom with a large bed, two nightstands, a floor lamp, a wardrobe, and a big window."

**Prompts for FPVSCORE.**   To facilitate consistent and goal-driven evaluations in FPVSCORE, we design structured prompts that include: (1) task-specific instructions for multi-scene comparison, (2) clearly defined evaluation criteria, and (3) standardized formatting requirements. These prompts guide the model to assess each scene in terms of semantic fidelity, spatial layout accuracy, and overall coherence, while also requiring concise justifications to support its ratings and enhance transparency.

> Task: Compare the room layout rationality of three methods, all generated from the same text description. From top to bottom, the video sequences display a 360-degree view of each method's generated scene. Decide which method performs best according to the criteria below.
> Text Description: {text_description}
>
> Instructions:
> 1. Semantic Correctness
> Does the generated layout accurately reflect the text description?
> Check whether all described objects are present and correctly represented.
> 2. Layout Correctness
> Is the room design physically plausible and functional?

Evaluate if the layout supports practical use, space efficiency, and proper object functionality. Consider object positions, orientations, and user convenience.
3. Overall Preference
Does the room layout look realistic and natural?
Consider the visual coherence and harmony of the scene.

Evaluation process:
Carefully examine the multi-view images of all three 3D scenes. Focus on one criterion at a time and make independent judgments for each.

Output format:
Provide a clear, concise analysis for each criterion. Avoid vague terms like "realistic" or "spacious." Instead, specify exact issues or strengths. For example:
- For Semantic Correctness, indicate which objects are missing or inaccurately depicted.
- For Layout Correctness, specify which objects are misplaced or poorly oriented, and explain how this impacts usability or functionality.
After the analyses, assign ranks (1–3) to each method per criterion (1 = best, 3 = worst).
Summarize your final ranking in the format: <rank for criterion 1> <rank for criterion 2> <rank for criterion 3>
for each method.

Example:
*Analysis:*
1. Semantic Correctness: The first one ...; The second one ...; The third one ...
2. Layout Correctness: The first one ...; The second one ...; The third one ...
3. Overall Preference: The first one ...; The second one ...; The third one ...

*Final answer:*
The first one: x x x
The second one: x x x
The third one: x x x
(where x denotes ranks 1–3)

(Please strictly follow the format above. Do not include extra symbols like **, quotation marks, or bullet points.)

**Prompts for Top-Down View Scores.**    Following the approach of Architect [60], we design targeted prompts to guide GPT-4o in evaluating room layouts based solely on top-down views. To ensure a fair comparison, the prompts also emphasize spatial structure, semantic fidelity, and functional usability, consistent with our own.

Task: Compare the room layout rationality of three methods, all generated from the same text description. The top-down views of the scenes produced by the three methods are presented from left to right. Identify which method performs best based on the criteria below.
Text Description: {text_description}

Instructions:
1. Semantic Correctness
Does the generated layout accurately reflect the text description?
Check whether all described objects are present and correctly represented.
2. Layout Correctness
Is the room design physically plausible and functional?
Evaluate if the layout supports practical use, space efficiency, and proper object functionality. Consider object positions, orientations, and user convenience.
3. Overall Preference
Does the room layout look realistic and natural?
Consider the visual coherence and harmony of the scene.

Provide only your final ranking of the three methods in the format below:

> *Final answer:*
> x x x
>
> (where x denotes ranks from 1 to 3)

## H    User Study Details

We conducted a thorough user study to evaluate the quality of the generated scenes, involving thirty participants. All participants took part voluntarily and received no compensation. At the start of the study, participants were given five minutes to read through the instructions, as illustrated in Fig. 13. An example evaluation page presented to the participants is shown in Fig. 14.

Figure 13: **User Study Instructions.** This page was shown to participants at the beginning of the study to explain the task, interface, and evaluation criteria.

## I    Limitations

VIPSCENE currently generates spatially coherent scene layouts, retrieving furniture from Obja-verse [7]. Although objects are richly annotated, some textures lack photorealistic quality. Future work will improve object quality by adopting advanced 3D generation techniques like text-to-3D and image-to-3D methods [62, 63, 66, 76], and by incorporating state-of-the-art physically-based rendering (PBR) techniques [5, 12, 16, 75] for realistic material representations and lighting. These improvements aim to enhance both the diversity and realism of generated scenes.

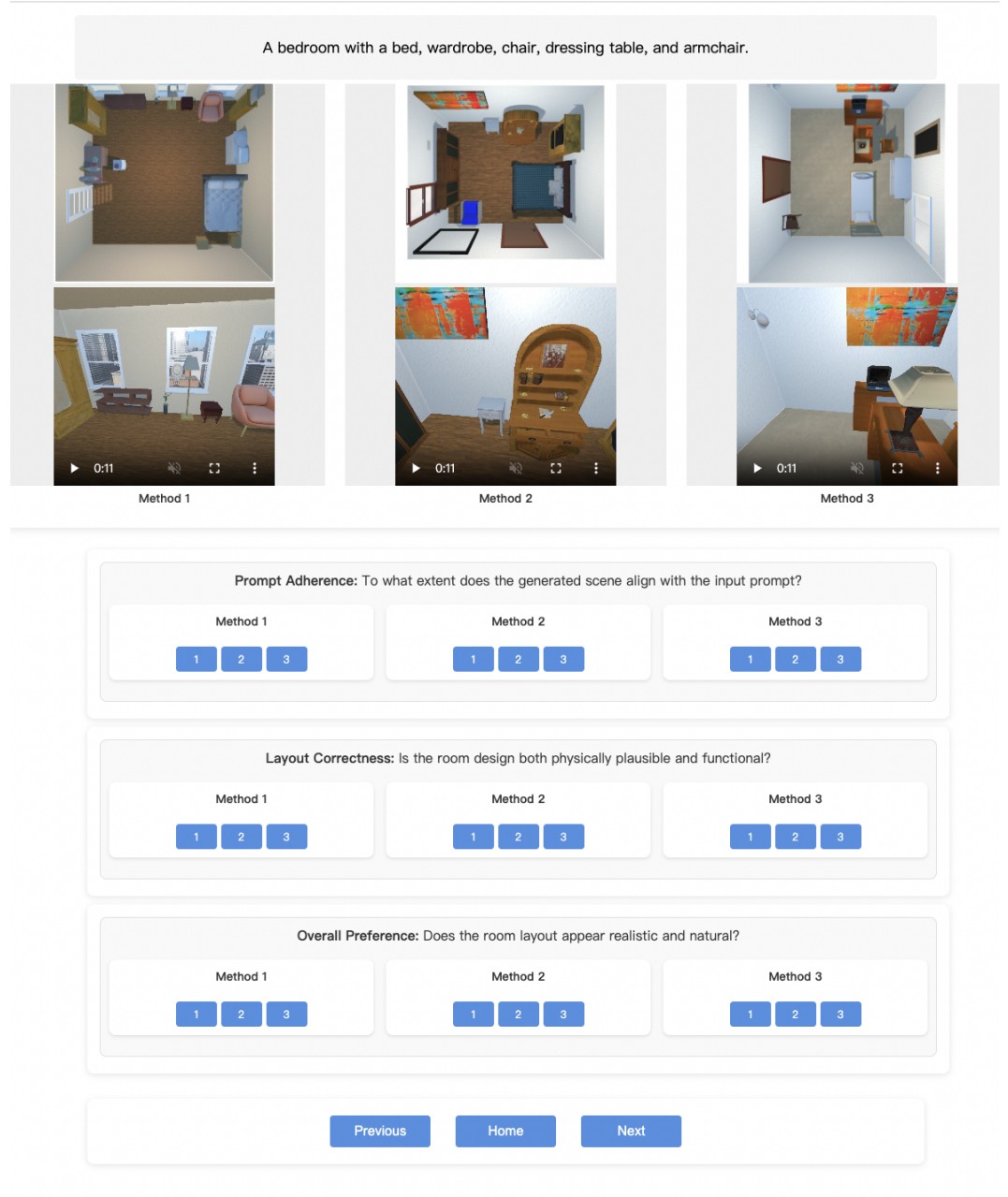

Figure 14: **Example Page.** Participants were shown a 360-degree video captured from the center of each scene, along with a top-down rendered image. This setup allowed them to evaluate both the global structure and fine details. Each scene was rated on a 3-point scale (1 = lowest, 3 = highest) across three criteria.

