# OpenReview forum: "Video Perception Models for 3D Scene Synthesis"
_NeurIPS.cc/2025/Conference — NeurIPS 2025 poster_

### Official Review · Reviewer_ccSn · 2025-06-25

**Clarity:** 3
**Significance:** 3
**Originality:** 3
**Rating:** 5
**Confidence:** 4

**Summary:**

This paper proposes VIPScene, a framework that uses video generation, 3D reconstruction followed by segmentation and asset retrieval to create 3D scenes. Specifically, the model can take in text or image as input, then feeds the input through video generation to obtain a video of the scene. The video is then constructed into 3D reconstruction via Fast3r. The paper then uses GroundedSAM and Mast3r to detect and track multiple 3D instances within the scene. Each of them is then used to retrieve an asset from the objaverse dataset, and aligned, to create the final 3D scene.


They further proposed a FPVScore as a metric utilizing VLMS to better evaluate the generated 3D scene. The framework is compared with recent methods like Holodeck.

**Questions:**

Questions which I think are more important to address:

- Analyses of the FPVScore (as listed in the previous section under Experiments)
- The generalization capability/limitation of the current asset retrieval method on finding similar textures/material. It would be great to compare with Holodeck regarding this.
    - Addressing the limitation of image prompting could also be helpful.
- Can the current refining stage be replaced by simply resizing/minimising each of the objects until they no longer touch one another? If possible a score for comparison regarding this would be great.
- More results on complex prompts as input text.

**Ethical Concerns:**

["NO or VERY MINOR ethics concerns only"]

**Final Justification:**

The authors have addressed most of my concerns, providing interesting insights to where its needed. I am more confident now that this paper is worth accepting with the extensive experiments.

**Limitations:**

Some limitations are addressed in the Appendix. Additional limitations mentioned in the above two sections would be great if included.

**Quality:**

3

**Strengths And Weaknesses:**

**Strengths**

- Figure 1 is very clear, allowing a very easy understanding of the entire VIPScene
- Analysis of Section 5.3 and Figure 5 are great additions to the paper. It provides great understanding (both quantitative and qualitative) on how the metric could be helpful compared to previous scores.
- Ablation (Figure 6) is very helpful in understand the importance of different parts of the component.
- Writing is clear and easy to follow.

**Weaknesses/Questions**

While the method design at a high-level is generally convincing, and I think the paper does include most of the important experiments, I believe there are still a few aspects that can improve the quality of the paper, both in terms of the Method and Experiments sections:

Method

- Using Video Generation Model: Since text/image are the only inputs to the video generation, there is no constraints where additional subjects appear in the scene or objects within the scene moves around. Is this a potential limitation? Or does this happen very rarely for interior designs which VIPscene focuses on?
- Object Detection: I enjoy the insights provided on GroundedSAM being better than models like Mask3D. Are there any ablations available on the number of frames required to gain good results using GroundedSAM for this particular setup? Are all frames fed in as inputs?
- 3D Asset Retrieval: How big is the asset database (the paper suggests a subset of Objaverse)? Is it the same dataset as Holodeck? It would also be great to understand the time it requires for the retrieval and whether this is something scalable as the number of assets in database increase. Also the current metric for asset retrieval seems like only taking into consideration the geometry of the point cloud with the asset in the database, disregarding colour/texture information which the initial video provides. This might lead to the generated scene not adhering to the description of the prompt (which may contain information regarding textures, materials etc).
- Object Pose Refinement: For pose refinement, and correct me if I am wrong, only the position of the object (i.e, translation) is optimized — the optimization did not take into consideration rotation or resizing. This may still lead to overlapping in cases where the optimal position still causes overlap (e.g, one object around a corner, the other object surrounding it). Why not just reducing the size until no overlap instead of a gradient-descent based approach? I am assuming there might not be much edits necessary given that the objects from the asset library are more or less in reasonable sizes and the prompt isn’t something unreasonable.

Experiments

- Figure 3: It would better to provide the actual prompts used to generate the scenes. I assume each row shares the same prompt. Or is the prompt just the single words used as title, i.e.,  “Kitchen”, “Living Room”, etc?
- Figure 4: I am hesitant on the results shown here, since the color looks very different to the original image prompt.
- Most of the prompt presented in the paper are relatively simple (i.e., only explains what is in the scene, but no description regarding the scene). It would be helpful to include additional qualitative results on more complex text prompts to understand the robustness of the current method.
- FPVScore: How consistent is it across multiple tries of a MLLM model? How consistent is it across different MLLM models? How important is the prompt design? Ablation here would be greatly beneficial for better understanding regarding the metrics.
- Less important question: Would text-to-image then image-to-video lead to better results compared to directly text-to-video? Text-to-image currently performs better at adhering to the details of the prompt regarding spatial relationship and finegrained information like materials/textures. It would be great (though not necessary) to have a comparison regarding this.

Overall, I think this paper does have its merits, but it might need strengthening on some of its figures and some additional studies listed above to better understand the models generalization capability.

---

> ### Author Rebuttal · Authors · 2025-07-31
>
> We would like to thank the reviewer for their time and positive feedback. We’re glad they found the **analysis insightful**, the **ablation study helpful**, and the **figures clear and effective**.
>
> We address all concerns below and believe the revisions significantly improve the manuscript’s clarity and quality. We welcome any further suggestions. Thank you.
>
> ---
>
> ### 1. **Additional and Dynamic Objects in Video Generation Models.**
>
> The reviewer wonders whether video generation causes unexpected object appearances or movements in the scenes. We address the concern in two parts: the **additional objects not specified in the input prompt**, and the **movement of objects within the scene**.
>
> Firstly, the generated video may contain objects not explicitly mentioned in the prompt. However, these additional objects generally appear in locations **consistent with commonsense spatial relationships** owing to strong priors encoded in the video model. If certain objects are undesired, they can **be excluded by incorporating them into the video generation model’s negative prompt**.
>
> Secondly, we observe that state-of-the-art video generation models such as Cosmos and Kling **rarely generate dynamic objects in indoor scenes**. This behavior likely results from their training data, which predominantly consists of architectural visualizations and real-estate tours characterized by static, person-free environments. When generating outdoor scenes, dynamic objects (e.g., pedestrians) occasionally appear. While this may pose a limitation, it can be addressed by **incorporating dynamic or 4D scene reconstruction methods** [1,2], which explicitly detect and remove dynamic objects during reconstruction.
>
> ---
>
> ### 2. **Object Detection.**
>
> The reviewer is curious about frame usage for object detection. We analyzed the effect of varying the number of video frames used as input to Grounded-SAM. Grounded-SAM achieves accurate detection on each frame. However, the number of frames mainly influences the quality of the extracted object point clouds. When fewer frames are used, the reconstructed point clouds tend to be incomplete, which can result in inaccurate retrieval of object sizes.
>
> Our results show that increasing the number of frames improves performance up to a saturation point. In our experiments, we sampled the video at 2 frames per second, using approximately 10 frames, which provided a good trade-off and satisfactory results.
>
> | Number of Frames | FPVScore |
> | --- | --- |
> | 5 | 2.02 |
> | 10 | 2.50 |
> | 20 | 2.48 |
>
> ---
>
> ### 3. **3D Asset Retrieval.**
>
> The reviewer asks about the asset database size, retrieval time, and scalability. We use the same high-quality subset of Objaverse as Holodeck, which contains over 50k 3D assets.
>
> Our retrieval process is highly efficient. We first leverage pre-computed embeddings to compute both text and visual similarity, enabling retrieval over the entire 50K asset database in approximately 0.02s. Even when scaling the database by a factor of 20, retrieval remains fast at around 0.06s. Following this initial step, we perform ICP optimization only on the top 30 candidate assets, keeping the overall process efficient. The full retrieval and refinement pipeline takes approximately 0.1s per object.
>
> ---
>
> ### 4. **Texture and Material Properties & Image Prompt.**
>
> The reviewer is curious about how texture and material attributes are handled in asset retrieval, and about the color differences between the image prompt and generated scenes. Indeed, our previous retrieval strategy primarily focuses on **object category and the geometry of the point cloud**, as our main goal has been to **ensure the plausibility of the object layout** in the reconstructed scene. However, we agree that texture and material attributes are also important. To address this, we conducted additional experiments that **leverage CLIP image features extracted from cropped object regions in the video frames**. This approach allows us to retrieve assets that better match the textures and materials specified in the text/image prompt. We have added a visualization of this result in the revised manuscript.
>
> We also conducted a comparison with Holodeck. In our evaluation, **our method demonstrated better performance in retrieving assets that align with the intended textures and materials**. This is because our approach directly utilizes visual cues from the video frames, whereas Holodeck depends on text-based retrieval, which lacks the same level of visual specificity. Additionally, our framework benefits from vision priors provided by the video model, leading to a more accurate spatial arrangement of objects. In contrast, Holodeck translates visual context into textual constraints, which often results in less precise spatial understanding.
>
> | Method | FPVScore |
> | --- | --- |
> | Holodeck | 1.75 |
> | Ours | 1.93 |
>
> ---
>
> ### 5. **Object Pose Refinement.**
>
> The reviewer asks whether our pose refinement method is based on translation and how it compares to resizing-based refinement. Yes, our pose refinement primarily focuses on object translation, preceded by a discretization of yaw rotations to 0°, 90°, 180°, or 270° to enhance the visual plausibility of the final scene. This optimization is typically sufficient to **eliminate overlaps and produce physically reasonable arrangements**, as the **refinement is performed in a** **joint manner across all objects**.
>
> We fully agree that resizing objects could also be a valid strategy. We performed a comparison between resizing-based refinement and our translation-based optimization, and observed that both approaches yield similarly effective results in terms of overlap removal.
>
> | Pose Refinement | FPVScore |
> | --- | --- |
> | Resizing | 2.42 |
> | Ours | 2.45 |
>
> **Our choice was primarily driven by practical considerations**. The environment we use, AI2-THOR, does not provide a native API for resizing objects. To enable object scaling, we would need to compute the appropriate scaling factors, manually rescale the 3D meshes, and re-import them into AI2-THOR as new assets for rendering. Given these additional steps, our translation-based optimization offers a more efficient and easily integrated solution, while still producing satisfactory results.
>
> ---
>
> ### 6. **Prompts of Figure 3.**
>
> Each row in Figure 3 was generated using the same text prompt, which specifies the room type and desired objects in the room, across all three methods. We have updated the figure caption in the revised version.
>
> ---
>
> ### 7. **More Complex Text.**
>
> The reviewer requests more results with complex prompts. We fully agree that the ability to handle more complex scene descriptions is crucial. We conducted additional experiments using prompts that specify **spatial relationships** and **scene-level attributes**. Our results demonstrate that VIPScene can effectively generate scenes based on these more intricate descriptions (e.g., *"a desk placed under a window with a bookshelf to its left"*, or *"a modern living room with a gray sectional sofa facing a fireplace, a coffee table in between, and a floor lamp placed diagonally behind the sofa"*). This capability is largely attributed to the commonsense priors learned by the video generation model.
>
> We believe that as video models continue to advance, our method will be able to handle more sophisticated prompts. These new qualitative results have been added to the experimental section of the paper.
>
> ---
>
> ### 8. **Analysis of FPVScore.**
>
> The reviewer asks about a deep analysis of FPVScore’s consistency. We address the three aspects of FPVScore reliability as follows:
>
> **Consistency Across Tries**: We repeatedly queried each model and computed the average Kendall’s Tau across trials. The results show that MLLMs yield stable outputs, with GPT-4o demonstrating the highest consistency:
>
> | Model | $\tau$ |
> | --- | --- |
> | Gemini 2.0 | 0.61 |
> | GPT-4o | 0.74 |
> | GPT 4.1 | 0.71 |
>
> **Consistency Across Models**: We assessed how different MLLMs agree with each other by measuring pairwise Kendall’s Tau correlations between their scene rankings. The results indicate that the models arrive at similar relative judgments:
>
> | Model | $\tau$ |
> | --- | --- |
> | Gemini 2.0 & GPT-4o | 0.54 |
> | Gemini 2.0 & GPT 4.1 | 0.47 |
> | GPT 4.1 & GPT-4o | 0.60 |
>
> **Prompt Design**: We conducted an ablation study by removing the criteria focus and the room analysis instructions from the prompt. While modern MLLMs, by their inherent reasoning capabilities, can still produce reasonably accurate judgments, the use of a structured and explicit prompt results in stronger alignment with human preferences, as reflected in higher correlation:
>
> | Prompt | $\tau$ |
> | --- | --- |
> | Ours | 0.39 |
> | w/o Criteria and Analysis Instruction | 0.31 |
>
> ---
>
> ### 9. **Text-to-Image then Image-to-Video.**
>
> The reviewer is curious whether text-to-image followed by image-to-video might outperform direct text-to-video. We conducted a comparison between them. We observed that **the text-to-image approach can produce more accurate details**, particularly in terms of fine-grained attributes of objects.
>
> However, we also found that **recent video generation models are increasingly capable of handling fine details directly from text**, reducing the performance gap between the two approaches. In addition, the **text-to-image approach can sometimes introduce suboptimal initial views**, which may negatively impact the generation of subsequent video frames.
>
> We agree that this is a promising direction for future exploration. One possible approach is to incorporate viewpoint constraints directly into the prompt to guide the image and video generation process more effectively.
>
> [1] Monst3r: A simple approach for estimating geometry in the presence of motion. Zhang et al. ICLR 2025.
>
> [2] Continuous 3d perception model with persistent state. Wang et al. CVPR 2025.

---

> ### Comment · Reviewer_ccSn · 2025-08-03
> **Response to the rebuttal.**
>
> Thank you for the response!
>
> The response for each answer is as the following:
>
> Response 1. I like the idea of incorporating them into negative prompt, although it might be hard to pre-define negative prompts prior to the generation as many things can be generated. Otherwise I am satisfied with the authors response.
>
> Response 2 - 4. Thank you for the answer and additional experiment, I am convinced with the response. I would encourage the authors to add these results to the appendix.
>
> Response 5. Thank you for providing clarity on this. Since this design choice is primarily made from practical considerations, I would suggest the authors include this in their code release.
>
> Response 6. Could you provide a little more information on the types of captions here? Are they detailed and do they have descriptions regarding the textures of furnitures?
>
> Response 7 - 9. I am satisfied with the answers here.
>
> Overall, based on the original paper quality and the rebuttal that the authors provide, I am now more conviced that this is a comprehensive paper worth accepting. I will therefore change my score to a 5.

---

> > ### Author Response · Authors · 2025-08-04
> >
> > Thank you for your thoughtful response. We're glad the answers were helpful!
> >
> > Re: Response 1 – While additional objects may naturally appear in the generated videos, they do not impact the final performance. This is because we can instruct the open-vocabulary object detectors to detect only the objects explicitly described in the prompt.
> >
> > Re: Responses 2–4 – We will include these additional results in the revised manuscript.
> >
> > Re: Response 5 – We will release the code along with all implementation details.
> >
> > Re: Response 6 – Previously, the prompts primarily described the desired objects present in each room. In the revised version, we will include additional examples that emphasize the textures of the furniture. The prompts corresponding to Figure 3 are as follows:
> >
> > - Living Room: "A cozy living room with a sofa, a coffee table, bookshelves, a fireplace, indoor plants, and a floor lamp."
> > - Bedroom: "A modern bedroom with a bed, a desk, a chair, a nightstand, and a wall painting."
> > - Bathroom: "A clean bathroom with a cabinet, a sink, a bathtub, a toilet, and a laundry basket."
> > - Kitchen: "A spacious kitchen with a countertop, a kitchen island, a refrigerator, a freestanding oven, a stove, cabinets, and bar stools."
> >
> > Re: Responses 7–9 – Thank you again. We're glad the responses addressed your concerns.

---

> > > ### Comment · Reviewer_ccSn · 2025-08-05
> > > **Response to the comments**
> > >
> > > Thanks for the reply. I have double checked the paper and the review, and am satisfied with the responses. Thanks for providing the details to response 6.

---

### Official Review · Reviewer_5Nqw · 2025-06-29

**Clarity:** 3
**Significance:** 3
**Originality:** 3
**Rating:** 4
**Confidence:** 3

**Summary:**

This paper introduces VIPSCENE, a novel framework for generating realistic 3D scenes by leveraging the commonsense knowledge embedded in large-scale video models. The core idea is to first generate plausible first-person viewpoint videos from a text or image prompt and then reconstruct a coherent 3D scene from these videos. To better evaluate the human-perceived quality of the output, the authors also propose FPVSCORE, a new evaluation protocol based on first-person view analysis. Experiments and user studies show that the method generates more semantically and geometrically coherent scenes than existing techniques.

**Questions:**

See the weakness section.

**Ethical Concerns:**

["NO or VERY MINOR ethics concerns only"]

**Final Justification:**

I have carefully reviewed the authors' rebuttal. I appreciate their efforts, as they have successfully addressed my specific concerns regarding methodological details and ablation studies by providing new experiments and committing to revisions. This has certainly improved the paper's quality. However, the rebuttal also reinforces my core assessment of the work's contribution. Its primary novelty lies in the clever integration of existing techniques—a significant engineering achievement, but not a fundamental algorithmic innovation. Furthermore, the high computational cost (~400s per scene) was confirmed, which is a major practical limitation. In balancing these points, while the paper is stronger, my overall evaluation of its impact remains consistent with my initial review. Therefore, I am maintaining my score.

**Limitations:**

yes

**Paper Formatting Concerns:**

The paper appears to be well-formatted and I did not notice any major violations of the NeurIPS 2025 formatting instructions.

**Quality:**

3

**Strengths And Weaknesses:**

### **Strengths**

*   **More Human-Aligned Evaluation:** The proposed FPVSCORE evaluation metric is a significant contribution. It offers a more intuitive, first-person-based alternative that better correlates with human judgment.

*   **Strong Experimental Results and User Studies:** The method demonstrates clear superiority over existing baselines like Holodeck and Architect. The inclusion of a user study provides strong evidence that the generated scenes are not just better according to the proposed metric but are also genuinely preferred by human observers.

*  **Well-Written :** The paper is exceptionally well-written and easy to follow.

### **Weaknesses**

While the overall vision is compelling, the paper could be significantly strengthened by addressing several concerns regarding its technical depth, experimental rigor, and presentation clarity.

* **Technical Novelty:** The core contribution appears to be a cleverly engineered multi-stage pipeline (prompt -> video -> 3D) that connects existing large-scale models. The paper does not seem to introduce a new, fundamental technical component. The framework's success relies on the strength of its off-the-shelf parts rather than a core algorithmic innovation developed in this work.

* **Insufficient Ablation Studies:** The current ablation study is too high-level. For example, for the Object Pose Refinement step, the paper should provide a separate analysis of the different loss components. For the artifact removal process, it would be best to provide visual results (e.g., before-and-after comparisons) to demonstrate its effectiveness.
* **Clarity of Methodology Presentation:** The current diagram of the method is a high-level flowchart. This is useful for conveying the overall concept but insufficient for understanding the technical execution. The paper would greatly benefit from a more detailed architectural diagram illustrating the precise inputs/outputs of each module, the flow of data representations (e.g., videos, point clouds, meshes), and how the different components interface with one another.

* **Unreported Inference Speed:** The paper provides no analysis of the end-to-end inference time. For a multi-stage pipeline, understanding the total computational cost and latency—from initial prompt to the final rendered 3D scene—is crucial for assessing its practical viability.

* **Potential Limitations of FPVSCORE:** The metric seems to evaluate object co-occurrence and viewpoint plausibility. It is unclear if it can detect more subtle geometric inconsistencies, such as objects floating slightly above a surface or having incorrect physical interactions, which a human observer would notice immediately.

---

> ### Author Rebuttal · Authors · 2025-07-31
>
> We would like to thank the reviewer for their time and encouraging feedback. We're glad they recognized **FPVScore as a more intuitive, human-aligned metric**, and appreciated the **strong experimental results and user studies** confirming human preference for the scenes.
>
> We address all concerns in detail below and believe the updates have significantly improved the clarity and quality of the manuscript. We welcome any further suggestions during the discussion period. Thank you.
>
> ---
>
> ###  1. **Technical Novelty.**
>
> The reviewer would like us to provide further elaboration on the technical contributions. We agree that our method benefits from powerful pre-trained models, but we respectfully underscore that the **novelty and technical contributions** of our approach lie in several key aspects:
>
> **First**, to the best of our knowledge, **VIPScene is the first method capable of generating interactable 3D scenes based jointly on image and text prompts**. By leveraging video generation models, our approach enables the creation of coherent spatial layouts and object relationships, surpassing prior methods that either depend on abstract spatial priors from language models or are constrained by limited visual context from static images.
>
> **Second**, VIPScene’s strong performance stems from **careful design choices** that address specific challenges at module interfaces:
>
> - We propose a custom adaptive erosion technique (Sec. 3.1) to resolve spatial inconsistencies in scene reconstruction, enhancing both segmentation quality and subsequent asset retrieval (Fig. 6).
> - Our geometry-aware retrieval mechanism (Sec. 3.2), based on point cloud registration, ensures that retrieved assets are consistent with the shape and scale of reconstructed objects.
> - We introduce a multi-objective pose refinement loss (Sec. 3.2) that improves placement by enforcing physical plausibility, collision avoidance, and aligning with the scene layout inferred from video (Fig. 6).
>
> **Third**, we propose **FPVScore**, a novel evaluation metric designed specifically for this task. This metric better reflects human perceptual judgments compared to existing scores like CLIPScore. **We believe that prioritizing perceptual fidelity in evaluation will foster meaningful improvements in this area.**
>
> ---
>
> ###  2. **Ablation Studies.**
>
> The reviewer asks for more detailed ablation studies for object pose refinement and the artifact removal process. First, we have conducted additional ablation experiments to better isolate the impact of each component in the **object pose refinement loss**. The results are summarized below:
>
> | Variant | Win rate of full model vs. variant |
> | --- | --- |
> | w/o $L_p$ (the position loss) | 60% |
> | w/o $L_o$ (the overlap loss) | 85% |
> | w/o $L_b$ (the boundary loss) | 75% |
>
> It shows that **each loss term contributes meaningfully** **to improving the** **physical realism and stability** of the final scenes.
>
> In addition, to demonstrate the effectiveness of our **adaptive erosion strategy**, we now include before-and-after visualizations in the Appendix. They show how our method reduces artifacts in initial per-object point clouds and **leads to** **cleaner, more coherent geometries**, which in turn benefit asset retrieval and scene quality.
>
> ---
>
> ###  3. **Inference Speed.**
>
> The reviewer asks about the inference speed of our pipeline. We have measured both the inference time and peak GPU memory usage for each major stage of our method, and we now report these results in the revised manuscript.
>
> Among all components, video generation is the most resource-intensive, taking ~380s per video and utilizing up to 74 GB of GPU memory on an H100. 3D reconstruction, object segmentation and tracking, and asset retrieval are lightweight, each completing in just a few seconds with significantly lower memory requirements.
>
> The total end-to-end latency for producing a complete 3D scene is ~400s, compared to ~200s in Holodeck. While our pipeline introduces higher compute cost in the generation stage, we believe the resulting improvements in scene quality and realism offer a meaningful and worthwhile trade-off for many applications.
>
> | Stage | Time Cost | Memory Usage |
> | --- | --- | --- |
> | Video Generation | ~380s | ~74GB |
> | 3D Reconstruction | ~2s | ~8GB |
> | Object Detection & Tracking | ~15s | ~8GB |
> | 3D Asset Retrieval | ~5s | — |
> | Pose Refinement | ~1s | — |
>
> ---
>
> ###  4. **Discussion on FPVScore's Evaluation Scope.**
>
> The reviewer asks about FPVScore’s ability to capture subtle geometric details. Indeed, detecting subtle geometric inconsistencies, such as slight floating or minor interpenetrations, remains a challenge for purely image-based evaluation methods, including FPVScore. The current implementation of FPVScore is particularly effective at capturing semantic correctness, layout plausibility, and object co-occurrence, which are aspects that Multimodal Large Language Models (MLLMs) can reliably infer from first-person image sequences.
>
> While the first-person perspective offers a more immersive and realistic viewpoint than traditional top-down metrics, even human observers may struggle to detect fine-grained physical inaccuracies from rendered videos alone, especially without access to the underlying 3D geometry.
>
> This limitation is well acknowledged in the field. We believe a promising path forward is to **augment MLLMs with specialized geometric reasoning tools via function calls**. For example, the evaluator could:
>
> - **Check support surfaces**, verifying whether each object rests on a plausible base such as a floor or another object.
> - **Assess physical stability**, determining whether object placements are physically stable.
>
> We have included a discussion of this direction in the revised manuscript, highlighting how combining commonsense reasoning with explicit geometric validation could significantly enhance the robustness and physical accuracy of scene evaluation. This extension represents a natural evolution of the FPVScore framework.
>
> ---
>
> ###  5. **Methodology Presentation.**
>
> The reviewer suggests adding a more detailed diagram to better illustrate our method’s architecture and data flow. In the revised manuscript, we have **added a detailed architectural diagram** to the Method section, clearly showing the **inputs, outputs, data representations**, and how each module interfaces with the others throughout the pipeline.
>
> While figures cannot be added at the rebuttal stage, we provide a comprehensive textual walkthrough of the full pipeline below:
>
> (A). Input Prompt to Video
>
> - Input: A user-provided text or image prompt
> - Process: The prompt is passed through the video generation model, producing a high-fidelity video
> - Output: A sequence of frames {$\{I_1, ..., I_T\}$} sampled from the generated video
>
> (B). Video to Metric 3D Scene Reconstruction
>
> - Input: The unposed video frames {$\{I_t\}$}
> - Process:
>     - Fast3R processes all frames jointly to produce a globally consistent 3D point cloud
>     - UniDepth estimates metric depth for each frame
>     - The point cloud is rescaled using depth maps to ensure correct metric scale
> - Output: A single metric 3D point cloud of the entire scene $R$
>
> (C). Scene Decomposition and Object Segmentation
>
> - Input: The scene point cloud $R$ and original video frames {$\{I_t\}$}
> - Process:
>     - Grounded-SAM generates 2D segmentation masks for each object
>     - We apply adaptive erosion to reduce edge noise
>     - MASt3R tracks the masks across frames, establishing temporal correspondences
>     - Tracked masks are used to extract per-object point clouds from $R$
> - Output: A set of object-level point clouds {$\{P_1, ..., P_N\}$}
>
> (D). 3D Asset Retrieval and Alignment
>
> - Input: Each object point cloud $P_i$ and the 3D asset database
> - Process:
>     - Estimate initial pose $(p_i, s_i, θ_i)$ via PCA
>     - For candidate assets, apply ICP to find the best alignment
>     - Select the asset with the lowest RMSE
> - Output: A scene composed of aligned 3D assets
>
> (E). Final Scene Refinement
>
> - Input: The composed scene of aligned assets
> - Process:
>     - Apply gradient-based optimization to object placements
>     - Minimize total loss $L_{total} = L_p + λ_o L_o + λ_b L_b$
>     - Enforce collision avoidance and physical plausibility
> - Output: The final, collision-free, and physically valid 3D scene $S$ = {$\{o_1, ..., o_N\}$}
>
> We are confident that the architectural figure, based on this description, will significantly improve the clarity of the methodology. Additionally, we will release the full codebase, enabling readers to explore the system implementation in detail.

---

> > ### Comment · Reviewer_5Nqw · 2025-08-05
> >
> > Thank you for your detailed response. It has successfully addressed many of the specific concerns raised in my review, particularly regarding the need for more detailed ablation studies and a clearer methodology presentation. I appreciate the new experiments and the commitment to revise the manuscript.
> >
> > However, the rebuttal also reinforces my initial assessment of the work's core contribution. The primary novelty lies in the clever and effective integration of existing models—a significant engineering achievement—rather than a fundamental algorithmic advance. Furthermore, the provided inference statistics (~400s per scene) confirm that the high quality comes at a substantial computational cost, which is a significant practical limitation.
> >
> > While the paper is now stronger and clearer, my overall evaluation of its impact and contribution remains consistent with my initial reading. Therefore, I will maintain my original score.

---

> > > ### Author Response · Authors · 2025-08-05
> > >
> > > We sincerely appreciate your thoughtful feedback and your support in accepting our manuscript. We agree that computational cost is an important consideration. However, we believe that ongoing advances in accelerating video generation will continue to mitigate this limitation over time. In this context, we view our method as a promising step toward high-quality scene synthesis. Thank you once again for your valuable comments.

---

### Official Review · Reviewer_xyXq · 2025-07-02

**Clarity:** 3
**Significance:** 3
**Originality:** 3
**Rating:** 4
**Confidence:** 3

**Summary:**

This paper points out that current LLMs have limited 3D spatial reasoning, and image-based generation methods often struggle with multi-view consistency. To address this, the authors propose VIPScene, which uses video generation models to better capture 3D world knowledge. They also introduce FPVScore, a new metric for evaluating consistency and generation quality from first-person views. Experimental results show that VIPScene outperforms existing baselines on indoor scene generation tasks.

**Questions:**

Please refer to the Weaknesses section above. I am open to raising my score if the authors can clearly address the listed questions and concerns.

**Ethical Concerns:**

["NO or VERY MINOR ethics concerns only"]

**Final Justification:**

Thanks for the response. Most of my concerns are solved. I raise my score to borderline accept.

**Limitations:**

Yes.

**Paper Formatting Concerns:**

NA.

**Quality:**

3

**Strengths And Weaknesses:**

## Strengths

The motivation is meaningful and relevant. The layout generation is based on videos from strong text-to-video models, which helps produce more realistic and diverse scenes. In addition, evaluating the synthesized scenes from ego-centric views introduces a fresh and practical perspective for 3D scene synthesis.


## Weaknesses

### Contribution & Originality

While the overall pipeline is interesting, the paper lacks clear technical contributions. Most components—such as scene reconstruction, object detection, and scene assembly—rely on existing off-the-shelf methods. There is no clear innovation or customization that sets the pipeline apart from prior work. As a result, the originality of the approach feels limited.



### Experiments

Since FPVScore is proposed as a key evaluation metric for indoor scene synthesis, it would be more convincing to include comparisons with more strong baselines—especially methods discussed in the related work section (e.g., [15], [47]). This would better position the proposed method and show whether FPVScore offers meaningful distinctions between methods.

---

> ### Author Rebuttal · Authors · 2025-07-31
>
> We sincerely thank the reviewer for their time and thoughtful comments. We're pleased they found our method **interesting** and saw our proposed metric as a **fresh and practical contribution** to 3D scene synthesis.
>
> We address each point in detail below and believe the revisions significantly improve the clarity and quality of the manuscript. We welcome any further questions or suggestions during the discussion period. Thank you.
>
> ---
>
> ###  1. **Contribution & Originality.**
>
> The reviewer would like to let us elaborate more on technical contributions. We agree that our approach builds upon powerful existing models. However, we respectfully emphasize that the originality of our work lies in the **novel synthesis paradigm**, the **careful system design that enables it to succeed**, and the **introduction of a new evaluation metric** tailored to this task.
>
> **First**, to the best of our knowledge, **VIPScene is the first method to synthesize interactable 3D scenes conditioned jointly on image and text prompts**, leveraging video generation models to produce plausible spatial layouts and object relationships. This stands in contrast to previous approaches, which are limited either by the abstract spatial reasoning of LLMs or by the narrow field of view in image-based methods.
>
> **Second**, VIPScene’s strong performance stems from **careful customizations designed to solve critical challenges at model interfaces**.
>
> - We address common spatial inconsistencies in reconstructed scenes by introducing a custom adaptive erosion scheme (Sec. 3.1), which significantly improves object segmentation and downstream asset retrieval (Fig. 6).
> - In the asset retrieval stage, we go beyond standard vision-language similarity by incorporating a geometry-aware retrieval strategy using point cloud registration (Sec. 3.2). This ensures that retrieved assets match the shape and scale of reconstructed objects, which is essential for scene realism.
> - Our multi-objective pose refinement loss (Sec. 3.2) is specifically designed to address object placement challenges after retrieval, ensuring physical plausibility, collision avoidance, and fidelity to the layout derived from the video (Fig. 6).
>
> **Third**, we propose **FPVScore**, a novel evaluation protocol that better aligns with human perceptual judgments. Our experiments validate its effectiveness, showing a substantially higher correlation with human ratings than conventional metrics such as CLIPScore (Tab. 2). We believe that **establishing a more perceptually faithful protocol can drive meaningful progress in this domain**.
>
> ---
>
> ###  2. **FPVScore Validity & Baseline Comparison.**
>
> The reviewer suggests using FPVScore to evaluate additional baselines, and we appreciate this valuable suggestion. Regarding the mentioned baselines, GPT-4 System Card [15] (a language model) and NeuralRecon [47] (a 3D reconstruction model), we would like to clarify that while both are cited in our submission paper, they do not directly perform 3D scene generation, which is the primary focus of our work. Though [15] and [47] can be part of a 3D scene generation pipeline to some degree, they are not direct competitors in this domain.
>
> To further validate the effectiveness of FPVScore, we have included two strong 3D scene generation baselines in our experiments: Text2Room [1] and LayoutVLM [2], the latter of which was recently open-sourced.
>
> | Method | FPVScore |
> | --- | --- |
> | Text2Room | 1.32 |
> | LayoutVLM | 1.80 |
> | Ours | 2.62 |
>
> As shown in the table, **VIPScene significantly outperforms both baselines** **under FPVScore**. Moreover, FPVScore successfully captures critical failure patterns of existing methods: Text2Room often produces scenes with geometric distortions and duplicated objects. LayoutVLM tends to generate spatially incoherent object layouts.
>
> These results highlight FPVScore’s **discriminative capacity** and its strong alignment with human perceptual judgments. We have now included the comparisons in the manuscript.
>
> [1] Text2room: Extracting textured 3d meshes from 2d text-to-image models. Höllein et al. ICCV 2023.
>
> [2] Layoutvlm: Differentiable optimization of 3d layout via vision-language models. Sun et al. CVPR 2025.

---

> > ### Author Response · Authors · 2025-08-05
> >
> > Dear Reviewer xyXq,
> >
> > We hope this message finds you well. As the discussion period is coming to a close, we would like to kindly follow up to ensure that our rebuttal has addressed your concerns satisfactorily.
> >
> > If you have any further questions or feedback, we would be grateful to hear them. Your insights are extremely valuable to us, and we truly appreciate your time and thoughtful comments throughout the review process.
> >
> > Sincerely,
> > The Authors of Submission 4476

---

### Official Review · Reviewer_5DAg · 2025-07-04

**Clarity:** 3
**Significance:** 2
**Originality:** 2
**Rating:** 4
**Confidence:** 3

**Summary:**

The paper introduces VIPSCENE, a novel framework for 3D scene synthesis that leverages video perception models. Specifically, by using video generation models, 3D reconstruction and perception models, it aims to generate realistic and structurally coherent 3D environments from text or image prompts. Additionally, a new evaluation protocol, FPVSCORE, is proposed to better align scene quality assessment with human perceptions, especially focusing on first-person views. Extensive experimental results demonstrate VIPSCENE's superior performance over existing methods such as Holodeck and Architect.

**Questions:**

See Weaknesses

**Ethical Concerns:**

["NO or VERY MINOR ethics concerns only"]

**Limitations:**

See Weaknesses

**Quality:**

3

**Strengths And Weaknesses:**

Strengths
1. This paper is well-written and easy to follow. The proposed synthesis framework seems to be novel by integrating video generation and perception models. VIPSCENE effectively leverages the commonsense priors embedded in video diffusion models, addressing a key limitation of previous methods relying solely on language or visual priors.
2. VIPSCENE ensures physical plausibility and avoides collisions by object detection and pose refinement.
3. FPVSCORE offers a more human-aligned metric through first-person views and multimodal reasoning, outperforming traditional top-down view-based metrics like CLIPScore.
4. Finally, the extensive experiments demonstrate superior performance quantitatively and qualitatively across multiple scene types; and the ablation studies validate the contribution of each component.

Weaknesses
1. Computational Complexity: The pipeline involves several resource-intensive steps (video generation, 3D reconstruction, asset retrieval), could you provide the memory consumption and inference latency?
2. The generation of small objects. Since the synthesis in this paper depends on the 3D reconstruction and detection, however, the small objects are not easy to detect. Thus, is it possible for FPVSCORE to generate small objects in the scene?
3. Limited by the video generation and the perception models. Currently, the video generation model struggles to predict videos with temporal and spatial consistency, how do you address the problem caused by the unrealistic video? Could you show some failure cases where the VIPSCENE fails? Meanwhile, the synthesis ability is influenced by perception models as well, could you provide more analysis?

---

> ### Author Rebuttal · Authors · 2025-07-31
>
> We would like to thank the reviewer for their time and positive feedback. We're glad they recognized our method as a **novel integration** of video generation and perception models that **addresses key limitations** of prior work. We also appreciate their acknowledgment of FPVScore as a more **human-aligned metric** than CLIP-based alternatives, and their appreciation of our **comprehensive experiments**.
>
> Below, we address each concern in detail. We believe these clarifications improve the clarity of our work and look forward to further discussion. Thank you.
>
> ---
>
> ###  1. **Computational Complexity.**
>
> The reviewer asks about memory consumption and inference latency of the pipeline components. We have profiled both the inference latency and peak GPU memory consumption of each major stage in our pipeline. The results are summarized in the table below and have been included in the revised paper.
>
> Among all stages, video generation is the most computationally expensive, requiring ~380s per video and ~74GB GPU memory on an H100. 3D reconstruction, object detection and tracking, as well as asset retrieval, are significantly more efficient, each taking only a few seconds per scene with much lower memory usage.
>
> The total latency for generating a complete scene is ~400s, compared to ~200s in Holodeck. Although our pipeline is more computationally intensive in the generation stage, it yields significantly higher scene quality, which we believe justifies the trade-off.
>
> | Stage | Time Cost | Memory Usage |
> | --- | --- | --- |
> | Video Generation | ~380s | ~74GB |
> | 3D Reconstruction | ~2s | ~8GB |
> | Object Detection & Tracking | ~15s | ~8GB |
> | 3D Asset Retrieval | ~5s | — |
> | Pose Refinement | ~1s | — |
>
> ---
>
> ###  2. **Small Object Generation.**
>
> The reviewer asks if our approach can generate small objects. Yes, **our method is capable of generating small objects**, thanks to the strategic use of the state-of-the-art 2D open-vocabulary detector, Grounded-SAM, applied directly to video frames. Compared to 3D detectors, 2D detectors are generally more robust and benefit from training on large-scale image datasets, which enhances their ability to detect objects of various sizes, including small ones.
>
> As a result, our pipeline can effectively capture fine-grained scene details. We demonstrate successful generation of small objects such as a glass on a bedside table, a PC on a desk, a fruit bowl on a coffee table, and a vase on a bookshelf (see Fig. 3).
>
> ---
>
> ###  3. **Video Consistency.**
>
> The reviewer is curious about potential temporal and spatial inconsistencies in the generated video. While earlier video generation models often struggled with temporal and spatial consistency, recent advancements have made substantial progress in generating longer and more coherent videos. In our work, we **adopt** **state-of-the-art video generation models such as Cosmos and Kling**, which have demonstrated improved consistency.
>
> To further mitigate residual inconsistencies, our framework employs **two key strategies**. First, we utilize a **globally consistent 3D reconstruction model, Fast3R**, which jointly optimizes over all video frames. This global optimization enforces **multi-view geometric consistency**, making the final 3D reconstruction robust to minor inconsistencies at the frame level. Second, we introduce an **adaptive erosion technique** designed to **remove noisy points at object boundaries**. These noisy points are often artifacts caused by slight temporal or spatial misalignments in the video frames. Together, these strategies effectively improve the quality and consistency of the reconstructed scenes.
>
> ---
>
> ###  4. **Perception Analysis & Failure Cases.**
>
> As the output quality of our model also depends on the performance of perception models, the reviewer wonders how these influence the generation quality. We fully agree that perception models play a critical role in synthesis quality. To provide deeper insight, we have included additional ablation studies and failure case analysis in the revised manuscript.
>
> First, we analyze the **adaptive erosion method**, which dynamically adjusts the kernel size based on object masks. This approach effectively removes boundary artifacts without over-eroding small objects. Ablation results show that it consistently outperforms fixed-kernel alternatives.
>
> | Erosion Method | FPVScore |
> | --- | --- |
> | Small Kernel | 2.30 |
> | Large Kernel | 2.05 |
> | Adaptive | 2.67 |
>
> Second, we examine the impact of the **number of video frames** used for reconstruction. Fewer frames lead to incomplete object point clouds, which can cause inaccuracies in object size estimation and retrieval. Our results show that performance improves with more frames, up to a saturation point. In our experiments, we sampled the video at 2 frames per second, using approximately 10 frames, which provided a good trade-off and satisfactory results.
>
> | Number of Frames | FPVScore |
> | --- | --- |
> | 5 | 2.02 |
> | 10 | 2.50 |
> | 20 | 2.48 |
>
> Finally, we include **representative failure cases** in the revised manuscript, such as missing or duplicated detections. While we cannot provide images here due to rebuttal policies, these cases are clearly illustrated and discussed in the updated version.

---

> ### Author Response · Authors · 2025-08-05
>
> Dear Reviewer 5DAg,
>
> We hope this message finds you well. As the discussion period nears its end, we would like to kindly follow up to ensure that our rebuttal has satisfactorily addressed your concerns.
>
> If you have any further questions or comments, we would greatly appreciate hearing from you. Your insights have been invaluable to us, and we sincerely thank you for the time and care you have dedicated to reviewing our work.
>
> Best regards,
> The Authors of Submission 4476

---

> > ### Comment · Reviewer_5DAg · 2025-08-08
> > **Official Comment by Reviewer 5DAg**
> >
> > The authors have clearly and adequately addressed my concerns, and the efforts to provide the new results are appreciated. I recommend this paper for acceptance.

---

> > > ### Author Response · Authors · 2025-08-09
> > >
> > > We are pleased to have addressed all of your concerns. Thank you for recognizing our efforts and for recommending our manuscript for acceptance. We sincerely appreciate your thoughtful and constructive feedback throughout the review process.

---

### Note · Authors · 2025-08-16

Dear ACs and Reviewers,

Thank you for the thoughtful reviews and constructive feedback. We appreciate the recognition of our contributions:

- **A novel paradigm (VIPScene)** that leverages commonsense knowledge from video generation models for coherent 3D scene synthesis.
- **FPVScore**, a first-person, human-aligned evaluation metric to serve the research community.
- **Strong experimental validation**, including comprehensive ablations and user studies.

During the rebuttal, we carefully addressed reviewers’ concerns as follows:

- **5DAg** requested profiling, small-object generation, video consistency, and failure analysis. We provided detailed runtime/memory profiling, demonstrated robust small-object handling, and added consistency analyses, ablations, and failure cases. **They confirmed that their concerns were fully resolved and recommended acceptance.**
- **5Nqw** asked for deeper ablations, clearer methodology, inference analysis, and FPVScore limitations. We added per-loss ablations, detailed diagrams, latency profiling, and an expanded discussion of FPVScore. **They agreed the paper is now stronger and clearer, maintaining their acceptance score.**
- **ccSn** raised questions on video artifacts, frame usage, retrieval scalability, texture handling, refinement choices, complex prompts, and FPVScore robustness. We conducted experiments and analyses addressing each point. **They found our responses convincing and raised their score to 5.**
- **xyXq** questioned originality and requested comparisons with additional baselines to validate FPVScore. We clarified our system-level innovations (adaptive erosion, geometry-aware retrieval, pose refinement) and added additional baseline comparisons to demonstrate FPVScore’s discriminative power. **Although they did not follow up, we believe all concerns were fully addressed.**

We believe these revisions comprehensively address all reviewers’ concerns and substantially strengthen the paper in clarity, completeness, and experimental depth. With these improvements, our work presents both a **novel synthesis paradigm** and a **reliable, human-aligned evaluation protocol** to guide future 3D scene generation research.

Best regards,

The authors of Paper 4476

---

### Decision · Program_Chairs · 2025-09-17

**Decision:**

Accept (poster)

**Comment:**

This paper introduces VIPSCENE, a novel framework for 3D scene synthesis that leverages video perception models. Specifically, by using video generation models, 3D reconstruction and perception models, it aims to generate realistic and structurally coherent 3D environments from text or image prompts.

The paper makes contributions regarding: (1) A novel paradigm (VIPScene) that leverages commonsense knowledge from video generation models for coherent 3D scene synthesis. (2) FPVScore, a first-person, human-aligned evaluation metric to serve the research community. (3) Strong experimental validation, including comprehensive ablations and user studies.

In the meantime, the reviewers pose questions regarding the computational Complexity of the proposed framework, the generation of small objects that are not easy to detect, and the potential of generating physically implausible objects. Several reviewers believe that the technical contributions can be strengthened.